# Pro-inflammatory cytokines mediate the epithelial-to-mesenchymal-like transition of pediatric posterior fossa ependymoma

Rachael G. Aubin [1,2,6], Emma C. Troisi[1,6], Javier Montelongo[1], Adam N. Alghalith [1,3], Maclean P. Nasrallah [4], Mariarita Santi[4,5] & Pablo G. Camara [1✉]

Pediatric ependymoma is a devastating brain cancer marked by its relapsing pattern and lack of effective chemotherapies. This shortage of treatments is due to limited knowledge about ependymoma tumorigenic mechanisms. By means of single-nucleus chromatin accessibility and gene expression profiling of posterior fossa primary tumors and distal metastases, we reveal key transcription factors and enhancers associated with the differentiation of ependymoma tumor cells into tumor-derived cell lineages and their transition into a mesenchymal-like state. We identify NFκB, AP-1, and MYC as mediators of this transition, and show that the gene expression profiles of tumor cells and infiltrating microglia are consistent with abundant pro-inflammatory signaling between these populations. In line with these results, both TGF-β1 and TNF-α induce the expression of mesenchymal genes on a patient-derived cell model, and TGF-β1 leads to an invasive phenotype. Altogether, these data suggest that tumor gliosis induced by inflammatory cytokines and oxidative stress underlies the mesenchymal phenotype of posterior fossa ependymoma.

[1] Department of Genetics and Institute for Biomedical Informatics, Perelman School of Medicine, University of Pennsylvania, Philadelphia, PA 19104, USA. [2] Department of Bioengineering, School of Engineering and Applied Sciences, University of Pennsylvania, Philadelphia, PA 19104, USA. [3] Department of Chemistry, School of Arts and Sciences, University of Pennsylvania, Philadelphia, PA 19104, USA. [4] Department of Pathology and Laboratory Medicine, Perelman School of Medicine, University of Pennsylvania, Philadelphia, PA 19104, USA. [5] Department of Pathology, Children's Hospital of Philadelphia, Philadelphia, PA 19104, USA. [6] These authors contributed equally: Rachael G. Aubin, Emma C. Troisi. ✉email: pcamara@pennmedicine.upenn.edu

Pediatric ependymoma is a significant therapeutic challenge despite the general improvement of adjuvant chemotherapies in the treatment of pediatric brain tumors during the past decades[1–4]. A roadblock to expanding treatment options is the current limited knowledge about the molecular mechanisms that underlie ependymoma tumorigenesis, progression, and metastasis. Genome-wide DNA-methylation profiling of ependymal tumors has led to their classification into nine groups associated with distinct anatomical location, age of diagnosis, prognosis, and ability to metastasize[5–7]. In young children, the most common group is posterior fossa ependymoma group A (PFA), representing ~70% of cases[5]. Based on their methylation profile, PFA tumors can be further divided into two subgroups, PFA-1 and PFA-2[8]. These subgroups express genes that are respectively active in the brainstem and the isthmic organizer during embryogenesis[8,9], indicating a possible origin in radial glia from these locations[10,11]. However, little is known about the gene-regulatory circuits associated with ependymoma tumor cells and their metastasis into other regions of the central nervous system.

Single-cell transcriptomic profiling of primary and recurrent pediatric ependymomas has uncovered the presence of cellular hierarchies within these tumors. Tumor-derived cell lineages originate from undifferentiated tumor cells and resemble developmental radial glia-derived populations of ependymal, neuronal progenitor, and glial progenitor cells[12,13]. These studies have also identified the presence of a cell population characterized by a mesenchymal-like gene expression signature that is enriched for NFκB target genes and is consistent with a response to hypoxia and cellular stress[12,13]. The mesenchymal gene expression signature of ependymoma has been associated with poor prognosis[12,14], similar to what has been observed in glioblastoma[15–17]. Although gliomas are not epithelial cancers, the activation of an epithelial-to-mesenchymal-transition-like (EMT-like) process in glioblastoma has been associated with tumor cell proliferation and migration[18–20]. Highly motile mesenchymal-like glioblastoma cells are typically associated with abundant microglia infiltration, suggesting the importance of cues from the tumor microenvironment for the EMT-like process[21]. Consistent with the presence of a similar mechanism in ependymoma, NFκB activation in PFA ependymoma tumor cells has been linked to their inflammatory microenvironment[22,23], and hypoxia has been shown to be essential for the growth and propagation of PFA tumor cells in culture[24].

Here, we investigate the specific gene-regulatory circuits that control the EMT-like process of posterior fossa ependymoma, and their relation to inflammation. For that purpose, we examine the chromatin accessibility and gene expression profiles of primary and metastatic posterior fossa ependymal tumors at single-cell resolution, and we identify key transcription factors and distal regulatory elements associated with the transition of ependymoma tumor cells into a mesenchymal-like state. We then study the effect of specific pro-inflammatory cues on ependymoma tumor cells using a patient-derived cell model. Our results suggest that the concerted action of TGF-β and NFκB signaling induced by the pro-inflammatory and oxidative tumor microenvironment mediates the EMT-like process of pediatric posterior fossa ependymoma. Based on these findings, we argue that prospective effective therapies for pediatric ependymoma will need to simultaneously target both proliferative neuroepithelial- and mesenchymal-like tumor cell populations.

## Results

### A single-nucleus transcriptomic atlas of primary and metastatic posterior fossa ependymoma.
To characterize the cell ecosystem of posterior fossa ependymoma, we considered a cohort of 46 pediatric tumors collected by the Children's Brain Tumor Network (CBTN). This cohort consisted of 41 primary tumors and recurrences in the posterior fossa and 5 spinal or cortical metastases derived from primary posterior fossa tumors (Supplementary Data 1). All patients with metastatic tumors had been treated with radiotherapy. We developed a classifier to disaggregate the cohort into known molecular groups based on the gene expression profile of the tumors and applied it to 44 tumors for which RNA-seq data was available from the CBTN[25] ("Methods"). The remaining two tumors were classified based on quantitative reverse transcription PCR (RT-qPCR) data. Consistent with other studies[5,8], most tumors in this cohort (42 out of 46) were classified as PFA, out of which 67% ($n = 28$) were identified as PFA-1 (Supplementary Data 1). We selected 5 primary, 1 progressive, and 3 metastatic grade 2/3 tumors from the cohort and performed massively parallel single-nucleus RNA sequencing[26] using archived flash-frozen tissue. Overall, we profiled the transcriptome of 25,349 nuclei (with an average of 2660 nuclei per sample and 544 detected genes per nucleus) and used these data to create a transcriptomic atlas of primary and metastatic posterior fossa ependymoma (Fig. 1). Since we were interested in cellular programs that are shared across the tumors, we consolidated the gene expression data of the nine tumors into a single latent space using the algorithm Harmony[27] and performed an integrative analysis (Fig. 1a and Supplementary Fig. 1). Upon clustering and differential expression analysis, we identified ten cell populations which we annotated according to the expression of known marker genes. These populations were consistent with those identified in previous single-cell RNA-seq studies of posterior fossa ependymoma[9,12,13], and included undifferentiated tumor cells, five tumor-derived cell populations (ependymal cells, astrocytes, neural progenitor cells (NPCs), intermediate progenitor cells (IPCs), and mesenchymal-like tumor cells (MLCs)), endothelial cells, mural cells, oligodendrocyte progenitor cells (OPCs), and microglia (Fig. 1a–c and Supplementary Data 2). To ensure that the results of our study were consistent across samples, we performed differential gene expression analysis individually in each sample and combined the $p$ values across the samples using Fisher's method. Tumor-derived ependymal cells were annotated based on the upregulation of cilium genes (gene ontology enrichment (GOE) adjusted $p$ value $= 5 \times 10^{-24}$), such as DNAAF1, DNAH3, and CFAP157 (Fig. 1b, c). In contrast, tumor-derived NPCs were characterized by the upregulation of genes related to chemical synaptic transmission (GOE adjusted $p$ value $= 1.4 \times 10^{-4}$) and neurogenesis (GOE adjusted $p$ value $= 1.5 \times 10^{-6}$), such as NRXN3, DGKG, and DAAM2 (Fig. 1b, c). The population of tumor-derived astrocytes was identified by the upregulation of canonical astrocytic markers, including GFAP, AQP4, GJA1, and S100B (Fig. 1b). Tumor-derived MLCs expressed mesenchymal glioma markers, such as CD44, VEGFA, CHI3L1, HIF1A, and CA9 (Fig. 1b, c), whereas tumor-derived IPCs upregulated cell cycle genes (GOE adjusted $p$ value $= 2 \times 10^{-97}$), such as TOP2A, MKI67, and ASPM (Fig. 1b). We were unable to assign a specific identity to the population of undifferentiated tumor cells since all the genes expressed by this population were also expressed by other tumor cells (Supplementary Data 2).

Each cell population was similarly represented across the nine tumors, except for the MLCs which were largely associated with two distal metastases (Fig. 1d, e and Supplementary Fig. 1). However, ependymal tumors present a substantial degree of spatial heterogeneity, and MLCs are localized in perivascular and perinecrotic regions[12]. Therefore, the observed cell population abundances in the single-nucleus data might not be fully representative of the overall abundances in the whole tumor.

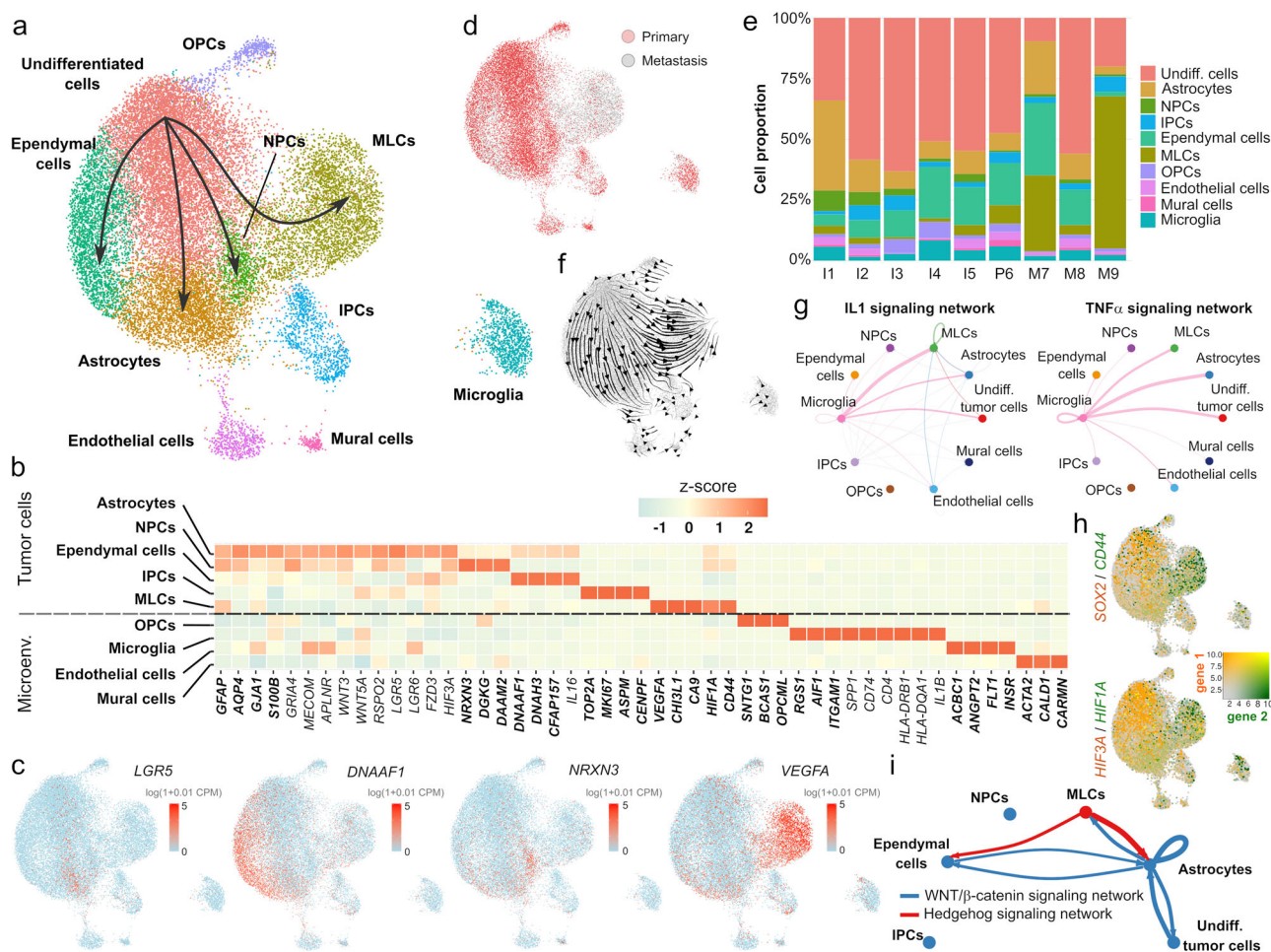

**Fig. 1 Single-nucleus RNA-seq uncovers the gene expression programs of primary and metastatic posterior fossa ependymoma. a** Single-nucleus RNA-seq data of 25,349 cells from nine tumors. The UMAP representation of the data is colored and annotated by the 10 cell populations identified. The four tumor-derived cell lineages are indicated by arrows. **b** Selected differentially expressed genes for tumor-derived and non-tumor cell populations, including marker genes (bold) that were used to annotate the cell populations, as well as differentially expressed genes discussed in the main text. Neuroepithelial-like tumor cell populations (undifferentiated cells, astrocytes, NPCs, and ependymal cells) express multiple genes coding for components of the WNT signaling pathway, whereas MLCs express high levels of hypoxia- and angiogenesis-related genes, and microglia express pro-inflammatory cytokines. **c** The UMAP representation is colored by the expression level of several differentially expressed genes that are associated with individual tumor-derived cell populations. **d** UMAP representation showing the origin (primary/metastasis) of each cell. The mesenchymal-like cell population is mostly associated with metastatic tumors. **e** Stacked bar chart depicting the proportions of each cell type in each tumor. **f** RNA velocity stream plot. The inferred cell differentiation trajectories originate in the undifferentiated tumor cell population and are consistent with the four lineages of tumor-derived cells indicated in **a**. **g** IL1 (top) and TNF-α (bottom) cell-to-cell signaling networks inferred by CellChat based on differentially expressed genes that code for ligands and receptor/co-receptors. The two networks show abundant pro-inflammatory signaling from microglia onto the tumor cells. **h** The UMAP representation of the single-nucleus RNA-seq data is colored by the expression level of some of the differentially expressed genes between neuroepithelial- and mesenchymal-like tumor cell populations. The UMAPs depict the expression of *SOX2* and *CD44* (top), and *HIF3A* and *HIF1A* (bottom). **i** Cell-to-cell WNT and hedgehog signaling networks inferred by CellChat indicating a switch between WNT signaling in neuroepithelial-like tumor cell populations and hedgehog signaling in MLCs. IPCs intermediate progenitor tumor cells, MLCs mesenchymal-like tumor cells, NPCs neural progenitor tumor cells, OPCs oligodendrocyte progenitor cells.

Tumor-derived cell populations were located adjacent to the cluster of undifferentiated tumor cells in the UMAP representation and had high expression of genes that were also expressed by the undifferentiated tumor cells (Fig. 1a and Supplementary Data 2). OPCs also appeared adjacent to the cluster of undifferentiated cells in this representation. However, it did not present CAs in subsequent single-nucleus ATAC-seq analyses (Supplementary Fig. 7), consistent with this cell population being mostly composed of non-transformed cells. We used scVelo[28] to infer RNA velocity trajectories based on the ratio of spliced and unspliced transcripts. The resulting trajectories were consistent with the presence of cell differentiation lineages from the undifferentiated tumor cells into the tumor-derived cell

populations (Fig. 1f), supporting the hypothesis that tumor-derived cell populations represent tumor cells that have undergone partial cell differentiation[9,12,13,29]. In agreement with previous results from bulk transcriptomics[30], numerous genes coding for components of the WNT signaling pathway were upregulated in the non-mesenchymal tumor cell populations, including *LGR5*, *LGR6*, *FZD3*, *RSPO2*, *WNT5A*, *WNT3*, and *DAAM2* (Fig. 1b, c and Supplementary Data 2). We collectively denoted these tumor cell populations (tumor-derived astrocytes, ependymal cells, and NPCs, and undifferentiated tumor cells) as neuroepithelial tumor cells. A gene-set enrichment analysis (GSEA) confirmed a significant enrichment for genes involved in the positive regulation of WNT signaling in this cell population

(normalized enrichment score = 3.05, *p* value = 0.04), suggesting the importance of WNT signaling in the proliferation and differentiation of ependymoma tumor cells.

**Posterior fossa ependymoma tumor cells and infiltrating microglia express pro-inflammatory cues.** We used the single-nucleus transcriptomic atlas to characterize the gene expression programs associated with the tumor microenvironment of posterior fossa ependymoma. Tumor-infiltrating microglia expressed MHC class II-related molecules characteristic of glioma-associated microglia[31], including *CD4*, *CD74*, and MHC-II α/β chains (Fig. 1b), reflecting a pro-inflammatory tumor micro-environment similar to what has been noted in spinal ependymoma[32]. We observed high expression levels of resident microglia marker genes and low expression of bone-marrow-derived macrophage markers[33] (Supplementary Fig. 2), although we cannot rule out the possibility that a substantial fraction of this cell population consists of bone-marrow-derived macrophages that have changed their gene expression profile upon recruitment to the tumor microenvironment[34].

In high-grade glioma, microglia are actively recruited to the tumor microenvironment through paracrine communication and chemotaxis with the tumor cells[34]. To investigate if a similar mechanism could take place in ependymoma, we systematically searched for differentially expressed genes that code for ligand and receptor/co-receptor pairs using the algorithm CellChat[35]. CellChat infers paracrine interactions using network analysis and categorizes them into signaling pathways based on existing databases. We focused our study on cytokine- and chemokine-mediated interactions between the tumor cells and the microenvironment. This analysis found the expression of several potent NFκB activating cytokines in microglia, including interleukin-1β (IL-1β), TNF-α, and TNFSF8, with the corresponding receptors being expressed in tumor cells (Fig. 1b, g, Supplementary Fig. 3, and Supplementary Data 3). Tumor cells also expressed high levels of genes coding for agonists of chemotactic receptors expressed in microglia. These chemotactic factors included IL-16, CD44, IL-6, and several colony-stimulating factor (CSF) and C-C motif ligand genes (Fig. 1b, Supplementary Fig. 3, and Supplementary Data 3). These molecules have been previously implicated in the chemotaxis and activation of microglia in high-grade glioma. Specifically, the hyaluronic acid receptor CD44 can interact with microglial osteopontin (*SPP1*) and is related to glioma tumor growth[36]; IL-6 has been associated with myeloid cell polarization in posterior fossa ependymoma[22]; and IL-16 is a potent chemotactic cytokine that binds to the CD4 receptor[37]. In addition, both microglia and tumor cells had high expression levels of the polyfunctional cytokine TGF-β (Supplementary Fig. 3 and Supplementary Data 3). Taken together, these results are consistent with the promotion of a microglia-rich microenvironment by posterior fossa ependymoma tumor cells and extend previous bulk transcriptomics results[22] by identifying MLCs as the main source for tumor-derived *IL-6*, *CSF1*, and *CD44* (Supplementary Fig. 3).

**Mesenchymal-like tumor cells are associated with abundant vascularization and microglia infiltration, and have elevated expression of NFκB target genes.** The presence of a microglia-rich microenvironment and the expression of inflammatory cytokines has been related to the mesenchymal transformation of high-grade glioma tumor stem cells under reduced oxygen levels and other cellular stresses[21,38]. To assess if a similar mechanism could underlie the generation of MLCs in ependymoma, we performed differential gene expression analysis between this population and the other tumor cells and characterized the gene

expression programs associated with the MLCs (Supplementary Data 4). Our analysis identified 1128 differentially expressed genes (false discovery rate (FDR) < 0.1), including a switch between *HIF3A* and *HIF1A* expression and downregulation of the neuroepithelial transcription factor gene *SOX2* (Fig. 1h and Supplementary Data 4). HIF3A is a negative regulator of the transcription factor HIF-1α, which mediates glycolytic and other major cellular responses to decreased oxygen levels[39]. The observed change in the expression of these genes thus suggests a possible role of HIF3A in maintaining the neuroepithelial phenotype of tumor cells. Consistent with these results, GSEA revealed a strong enrichment in the MLCs for programs that are characteristic of reactive gliosis during brain injury and neuroinflammation[40,41], including angiongenesis, hypoxia, glycolysis, NFκB, mTORC1, and sonic hedgehog signaling (Fig. 2a, b). The mesenchymal gene expression signature of glioblastoma[17,42] was also enriched in the MLC population (Fig. 2a), as previously reported[12,13]. The results from the ligand-receptor gene expression analysis were consistent with the activation of these pathways in the MLCs (Fig. 1g, i, Supplementary Fig. 3, and Supplementary Data 3). In addition, they showed the presence of abundant autocrine TGF-β, fibroblast growth factor (FGF), and hepatocyte growth factor signaling, as well as paracrine TGF-β and TNF-like weak inducer of apoptosis (TWEAK) signaling[43] from tumor-derived astrocytes, in this cell population (Supplementary Fig. 3 and Supplementary Data 3).

To further support these observations, we used the algorithm CIBERSORTx[44] to infer the abundance of the cell populations identified in the single-nucleus RNA-seq in each of the tumors from the large CBTN cohort profiled with bulk RNA-seq (Supplementary Fig. 4). CIBERSORTx uses support vector regression to estimate the relative proportion of each cell population in the bulk tissue based on input reference gene expression profiles. Consistent with the hypothesis that MLCs promote and sustain an inflammatory microenvironment through IL-6, CSF, and CD44 signaling, which results in the accumulation of tumor-infiltrating microglia and the activation of NFκB signaling in the tumor cells, the inferred abundance of MLCs was strongly correlated with the expression of IL-6/STAT3, TNF-α, TGF-β, mTORC1, and NFκB signaling gene sets in this cohort, as well as with the abundance of microglia (Spearman's correlation between MLC and microglia abundance *r* = 0.70, FDR < 0.001). The abundance of MLCs was also correlated with the expression of hypoxia, angiogenesis, glycolysis, and reactive oxygen species programs (Fig. 2c), and with the amount of vasculature (Spearman's correlation between MLC and mural cell abundance *r* = 0.85, FDR < 0.001). Tumors with a high abundance of MLCs were mostly classified as PFA-1 (Fig. 2c, PFA-1 enrichment score = 0.57, *p* value = 0.002). The GSEA score for sonic hedgehog signaling was also correlated with the abundance of MLCs (Fig. 2c, Spearman's *r* = 0.47, FDR < 0.01). In contrast, the abundance of MLCs presented an inverse relation with the expression of neuroepithelial-like marker genes identified in the single-nucleus RNA-seq analysis (Figs. 2c and 1b). These correlations were confirmed in an independent cohort of 83 pediatric posterior fossa tumors[5] (Supplementary Figs. 4 and 5). Altogether, these results are consistent with the association of MLCs with vascularization and microglia infiltration, and the activation of NFκB in these cells by pro-inflammatory cytokines from the microenvironment and the tumor.

**Single-cell chromatin accessibility profiling enables the systematic study of primary and metastatic posterior fossa ependymoma gene regulatory circuits.** To characterize the regulatory programs involved in the differentiation and EMT-like process of

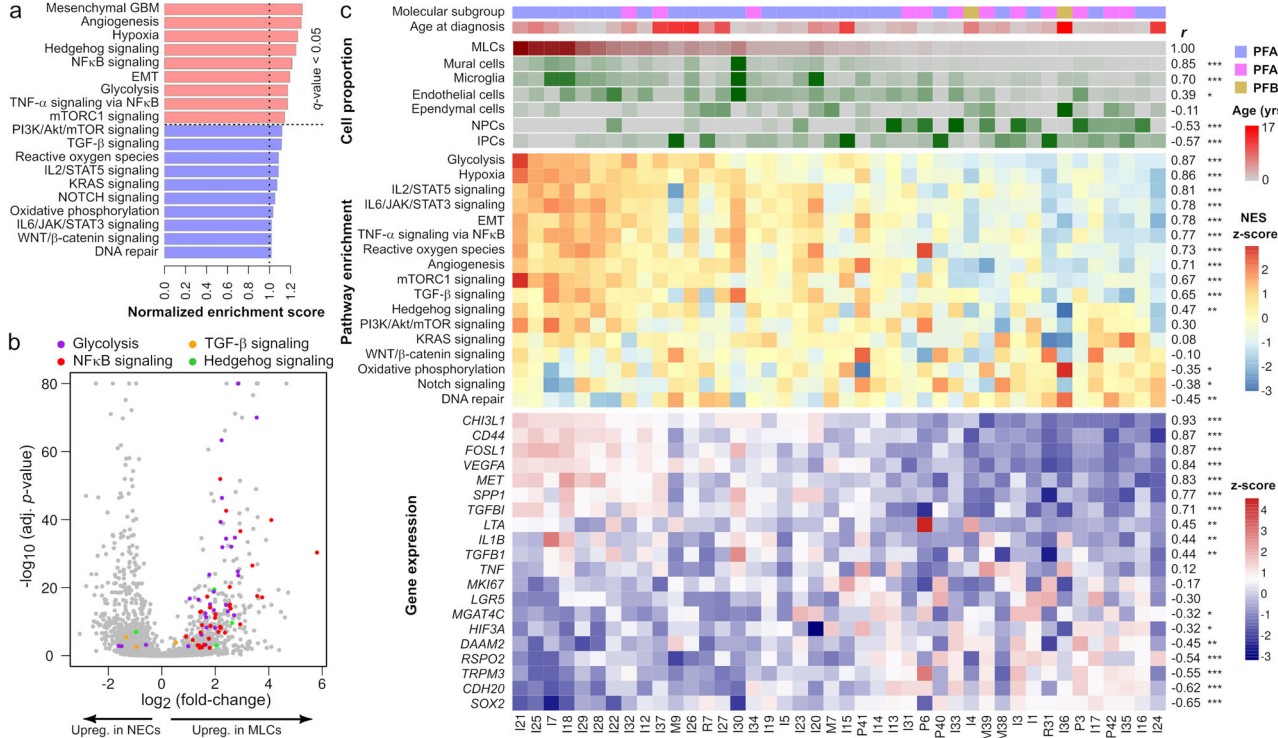

**Fig. 2 The mesenchymal-like tumor cell population of posterior fossa ependymoma is associated with hypoxia, angiogenesis, glycolysis, and NFκB signaling gene expression programs. a** GSEA in the MLC population showing enrichment for hypoxia, angiogenesis, glycolysis, NFκB, and other signaling pathways in this population. Normalized enrichment scores larger or smaller than 1 indicate an enrichment or depletion of the gene set in the MLC population, respectively. **b** Volcano plot showing differentially expressed genes between neuroepithelial- and mesenchymal-like tumor cells, where genes belonging to glycolysis, NFκB signaling, TGF-β signaling, and hedgehog gene sets are indicated (Fisher's method combined *p* value, adjusted for multiple hypothesis testing using Benjamini-Hochberg procedure). **c** The single-nucleus gene expression data were used to infer cell population abundances in a cohort of 42 pediatric posterior fossa ependymal tumors profiled with bulk RNA-seq by CBTN. Tumors are arranged from left to right by decreasing inferred abundance of MLCs. From top to bottom, the molecular subgroup, age of diagnosis, inferred abundance of distinct cell populations, GSEA scores of various representative signaling pathways, and expression of representative genes are shown. On the right side, the Spearman's correlation coefficient of each of these features with the inferred abundance of MLCs and its level of significance are indicated (two-sided test of association; *FDR ≤ 0.05, **FDR ≤ 0.01, ***FDR ≤ 0.001). Numeric *q* values are provided in the Source Data file. IPCs intermediate progenitor tumor cells, MLCs mesenchymal-like tumor cells, NPCs neural progenitor tumor cells, NECs neuroepithelial-like tumor cells.

posterior fossa ependymoma tumor cells, we performed single-nucleus ATAC-seq on 6 of the primary and metastatic tumors profiled with single-nucleus RNA-seq. We generated a high-resolution single-cell chromatin accessibility atlas consisting of 14,461 nuclei and 229,286 accessible peaks (2410 nuclei per sample, 8122 fragments per nucleus, and 4.5 transcription start site (TSS) enrichment score per nucleus on average) that passed the quality control metrics ("Methods"). To identify programs that are shared across the tumors, we built a consolidated representation of the six tumors, and clustered the cells based on their chromatin accessibility profile. We then computed the activity of each gene based on the number of accessible peaks that overlapped with the gene and used the marker genes identified in the single-nucleus RNA-seq data (Fig. 1b) to identify and annotate those cell populations (Supplementary Fig. 6). In total, we identified 11 distinct cell populations, including the populations identified in the single-nucleus RNA-seq data (except for the IPC population) as well as T cells and mature oligodendrocytes (Fig. 3a). The abundance of each cell population in each sample was moderately correlated with the observed abundance of the same population in the single-nucleus RNA-seq data of the same tumor (Figs. 3b and 1e, Pearson's *r* = 0.56, *p* value < 10⁻⁴), likely due to the large degree of spatial heterogeneity of posterior fossa ependymoma[12]. We used Copy-scAT[45] to infer large-scale chromosomal aberrations (CAs) from the single-nucleus

ATAC-seq data. Copy-scAT aggregates DNA fragments into uniform bins along the genome to identify cell populations with significant deviations in the abundance of fragments which might be representative of CAs. This analysis identified CAs in two of the tumors (I4 and M8), including a loss of chromosome 10 in both samples (Fig. 3d and Supplementary Fig. 7). In each tumor, MLCs had the same CAs as the other tumor cell populations, suggesting a common origin. We did not detect CAs in the non-tumor cell populations, consistent with our annotation of these populations (Fig. 3d and Supplementary Fig. 7).

To uncover the transcriptional regulators associated with tumor cells, we used the algorithm chromVAR[46] to infer the activity of transcription factors in each cell population based on the genome-wide accessibility of the corresponding binding motif. In total, we identified 554 transcription factors that were differentially active in one or more cell populations (Fig. 3e, f and Supplementary Data 5, Wilcoxon rank-sum test, FDR < 0.1). The inferred activity of the transcription factors MEIS1/2 was associated with both the mesenchymal- and the neuroepithelial-like undifferentiated tumor cell populations (Fig. 3f), thus representing a potential candidate for the development of therapies that simultaneously target both populations[47]. In addition, the activation of several transcription factors was associated with the differentiation of ependymoma tumor cells into tumor-derived cell populations. These transcription factors

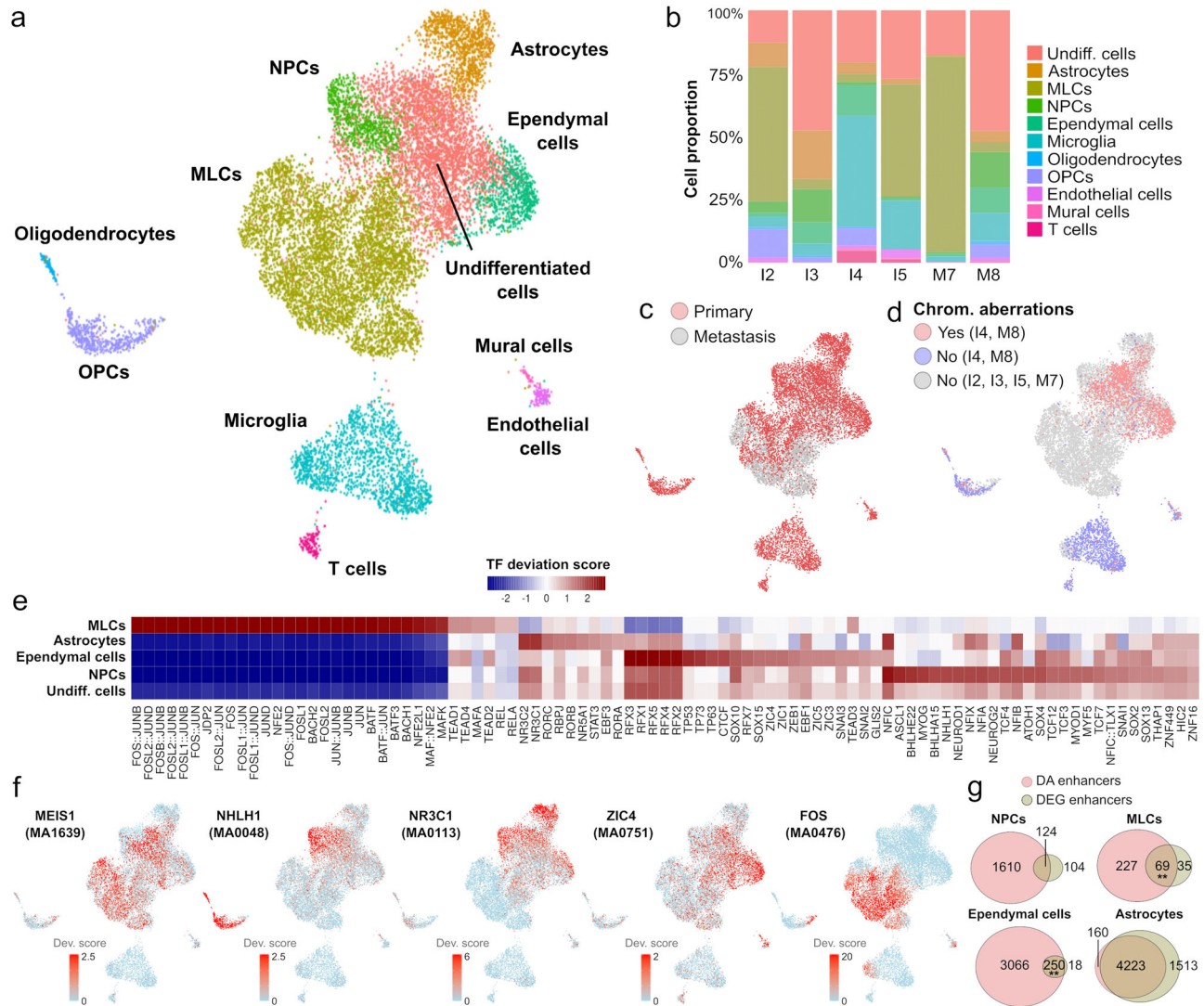

**Fig. 3 Single-nucleus ATAC-seq of primary and metastatic ependymal tumors uncovers transcription factors and enhancers associated with the differentiation and EMT of posterior fossa ependymoma. a** Single-nucleus ATAC-seq data of 14,461 cells from six primary and metastatic posterior fossa ependymal tumors. The UMAP representation of the data is colored and annotated by the 11 cell populations identified. **b** Stacked bar chart showing the proportion of each cell type in each of the six tumors. The UMAP representation is labeled according to the origin (primary/metastasis) of each cell (**c**), and the presence/absence of chromosomal aberrations as inferred from the single-nucleus ATAC-seq data (**d**). **e** Transcription factor (TF) binding motif accessibility score for TFs with differentially accessible binding motifs (Wilcoxon rank-sum test; FDR < 0.01) in at least one of the tumor cell populations and z-score fold-change $\Delta z \geq 1$. **f** The UMAP representation is colored by the TF binding motif accessibility score of five representative TFs that are significantly associated with individual tumor cell populations and are discussed in the main text. The JASPER database ID for the corresponding motif is indicated in parenthesis. **g** Venn diagrams showing the overlap between differentially accessible (DA) enhancers and enhancers associated with differentially expressed genes (DEGs) in tumor-derived cell populations. The number of enhancers in each class and the level of significance for the association are indicated (two-sided Fisher's exact test; **p value ≤ 0.01). Numeric p values for MLCs and ependymal cells are 0.0016 and 0.0075, respectively. MLCs mesenchymal-like tumor cells, NPCs neural progenitor tumor cells, OPCs oligodendrocyte progenitor cells.

included ZIC3/4/5 and the motile ciliogenesis regulators RFX1/2/3/4/5[48], which were associated with ependymal tumor cell differentiation, the glucocorticoid and mineralocorticoid receptors NR3C1/2, which were associated with astrocytic tumor cell differentiation, and the pro-neural transcription factors ASCL1 and NHLH1[49], which were associated with the differentiation into tumor-derived NPCs (Fig. 3e, f and Supplementary Data 5). Our results were also consistent with the histone 3 lysine 27 acetylation chromatin immunoprecipitation and sequencing (H3K27ac ChIP-seq) marks of pediatric ependymoma[50]. In particular, 41% (n = 229) of the differentially active transcription factors identified in our analysis were identified as core transcription factors based on bulk-level H3K27ac marks, of

which 82% (n = 187) were associated with tumor cell populations whereas the rest were associated with the microenvironment.

To infer the regulatory logic associated with the gene expression programs of the tumor cell populations, we identified differentially accessible peaks in each cell population that overlapped gene enhancers annotated in the GeneHancer database[51]. In total, we identified 8252 differentially accessible enhancers associated with at least one tumor cell population (Supplementary Data 6, Fisher's exact test FDR < 0.05). As expected, the genes regulated by these enhancers significantly overlapped with the differentially expressed genes found in the single-nucleus RNA-seq analysis of the same cell populations (Fig. 3g). Taken together, this multi-modal single-nucleus atlas

provides a unique resource for the study of gene regulatory circuits involved in the tumorigenesis and progression of pediatric posterior fossa ependymoma.

**Neuroepithelial transcription factors are inhibited, and NFκB and AP-1 complexes are activated, during the EMT of posterior fossa ependymoma.** We used the single-nucleus atlas and bulk gene expression data to identify key regulatory elements associated with the EMT-like process of posterior fossa ependymoma. For that purpose, we adopted a conservative approach and focused on transcription factors whose expression is significantly correlated or anti-correlated (Spearman's correlation $p$ value < 0.05) with the abundance of MLCs according to the bulk data and have differentially accessible binding motifs (Wilcoxon rank-sum test, FDR < 0.05) across the EMT-like transition according to the single-nucleus data. This approach identified 42 and 19 transcription factors that were significantly associated with the neuroepithelial- or mesenchymal-like tumor cell populations, respectively (Fig. 4a and Supplementary Data 7). Transcription factors that were active in the neuroepithelial-like tumor cells included transcription factors expressed in the neuroepithelium of the brainstem and isthmic organizer during mouse embryogenesis according to RNA in situ hybridization data from the Allen Developing Mouse Brain Atlas (Fig. 4a, b). These included SOX2, the nuclear factors NFIB and NFIX, and the regulatory factor X transcription factors RFX2/3, which play critical roles in the maintenance and differentiation of neuroepithelial stem cells during neurodevelopment[52–54]. Neuroepithelial transcription factors were inactive in the MLCs. Instead, this cell population was characterized by the activation of the NFκB and AP-1 transcription factor complexes and MYC (Fig. 4a). These transcription factors are activated by a variety of pro-inflammatory stimuli and oxidative stresses and play essential pro-oncogenic roles in multiple cancers[55–57]. In particular, both MYC and AP-1 mediate the oncogenic functions of ERK and p38/JNK signaling cascades[58–60], and these signaling pathways, as well as NFκB, can trigger and modulate the EMT in epithelial cancers[56,61,62]. Consistent with the accessibility of NFκB binding motifs, the promoter of LDOC1, a negative regulator of NFκB in ependymoma and other cancers[23,63,64], was only accessible in neuroepithelial-like tumor cells (Fig. 4c). Members of the MAF/BACH and YAP/TAZ complexes, which have been respectively implicated in the metabolism and metastatic potential of cancer cells[65] and the mesenchymal differentiation of glioblastoma[66], were also active in the MLCs (Fig. 4a). To elucidate potential regulatory interactions between the NFκB, AP-1, MYC, and MAF/BACH signaling pathways during the EMT of posterior fossa ependymoma, we identified differentially accessible binding motifs of these transcription factor complexes in the loci of genes that code for elements of these complexes. This analysis identified a substantial amount of crosstalk between transcription factors from these pathways (Fig. 4d) and highlighted activating transcription factor 3 (ATF3) as a potential integrator of signals from the NFκB, AP-1, and MAF/BACH signaling pathways during the EMT-like process of ependymoma tumor cells (Fig. 4d, e).

**Ependymoma mesenchymal-like tumor cells consist of multiple cell subpopulations with distinct transcriptomic profile and signaling activity.** We next investigated cell variability within the MLCs. We first used a spectral graph approach[67] to characterize the transcriptional heterogeneity within this cell population. This approach identified 4714 genes with a significant pattern of expression (FDR < 0.1) and revealed multiple subpopulations of MLCs marked by the expression of SERPINE1, MET, CHI3L1, ITGA10, GPC5, RGS2, and fibronectin 1 (FN1), among other

genes (Fig. 4f and Supplementary Data 8). An analysis of the MLCs by clustering and differential gene expression analysis identified 4 subpopulations with distinct gene expression profiles (Supplementary Fig. 8 and Supplementary Data 9). These subpopulations included a transitional population characterized by the co-expression of neuroepithelial (e.g., RFX3) and mesenchymal genes (e.g., VEGFA and CD44), a metabolic population characterized by the expression of hypoxia (GOE adjusted $p$ value = $9 \times 10^{-6}$), histone demethylation (GOE adjusted $p$ value = $4 \times 10^{-4}$), and glycolysis genes (GOE adjusted $p$ value = $8 \times 10^{-4}$), a population expressing high levels of angiogenesis genes (GOE adjusted $p$ value = $3 \times 10^{-5}$) as well as FN1, CHI3L1, and SERPINE1, and a small population with expression of the gene GPC5, which has been linked to cell proliferation via sonic hedgehog signaling in mesenchyme-derived tumors[68] (Fig. 4f). These analyses also revealed the upregulation of several genes associated with multi-drug and radiation therapy resistance. In particular, ABCC3, which codes for Multi-drug Resistance-Associated Protein 3 and is associated with chemotherapeutic resistance in several cancers[69,70], was expressed at high levels in angiogenic MLCs (Supplementary Fig. 9a). The expression of PDK1, which leads to HIF-1α-mediated radiation resistance[71], was upregulated in metabolic MLCs (Supplementary Fig. 9a). In addition, MLCs expressed high levels of the NF-κB target gene SOD2 (Supplementary Fig. 9a), which mediates radiation resistance through oxidative stress modulation[72,73]. Consistent with these results, the expression of all of these genes in the larger CBTN and Heidelberg cohorts was strongly correlated with the inferred abundance of MLCs in each patient (Supplementary Fig. 9b, c).

We followed the same approach to investigate the heterogeneity of the MLC population in the single-nucleus ATAC-seq data. This identified a substantial amount of heterogeneity in the activity of several transcription factors within the MLC population, including EGR1, ATF3, MYC, and TEAD3/4 (Fig. 4b, g and Supplementary Data 10). The patterns of activity for these transcription factors were consistent with their expression profile in the single-nucleus RNA-seq data. For example, the gene expression level of ATF3 and the accessibility of its binding motif were downregulated in angiogenic MLCs, which had high gene expression levels and binding motif accessibility of the transcription factor EGR1 (Fig. 4g). On the other hand, MYC and ATF3 were upregulated in metabolic MLCs. Taken together, these results indicate that the modulation of the AP-1, MYC, and NFκB signaling pathways, possibly due to differences in the local tumor microenvironment, is associated with substantial heterogeneity within the MLC population.

**TGF-β1 and TNF-α respectively induce and modulate the EMT in a patient-derived PFA cell model.** To directly investigate the effect of pro-inflammatory cytokines in posterior fossa ependymoma tumor cells, we considered the cell model EPD-210FHTC[74]. This early passage cell line was derived from a M0 stage PFA ependymoma recurrence in a 10 year-old patient, and it recapitulates the gene expression, DNA methylation, copy number, and mutation profiles of the original tumor[74]. Since the results of our single-nucleus data analysis indicated abundant TNF-α/NFκB and TGF-β signaling in the MLC population and expression of these two cytokines in tumor-infiltrating microglia (Figs. 2 and 4 and Supplementary Fig. 3), we focused our analysis on the effect of these cytokines. We cultured the cells for 5 days in neural stem cell basal medium supplemented with recombinant FGF (20 ng/ml) and epidermal growth factor (EGF) (20 ng/ml), in the presence or absence of human TGF-β1 (4 ng/ml) and/or TNF-α (10 ng/ml) (Fig. 5a). In absence of TGF-β1, cells formed

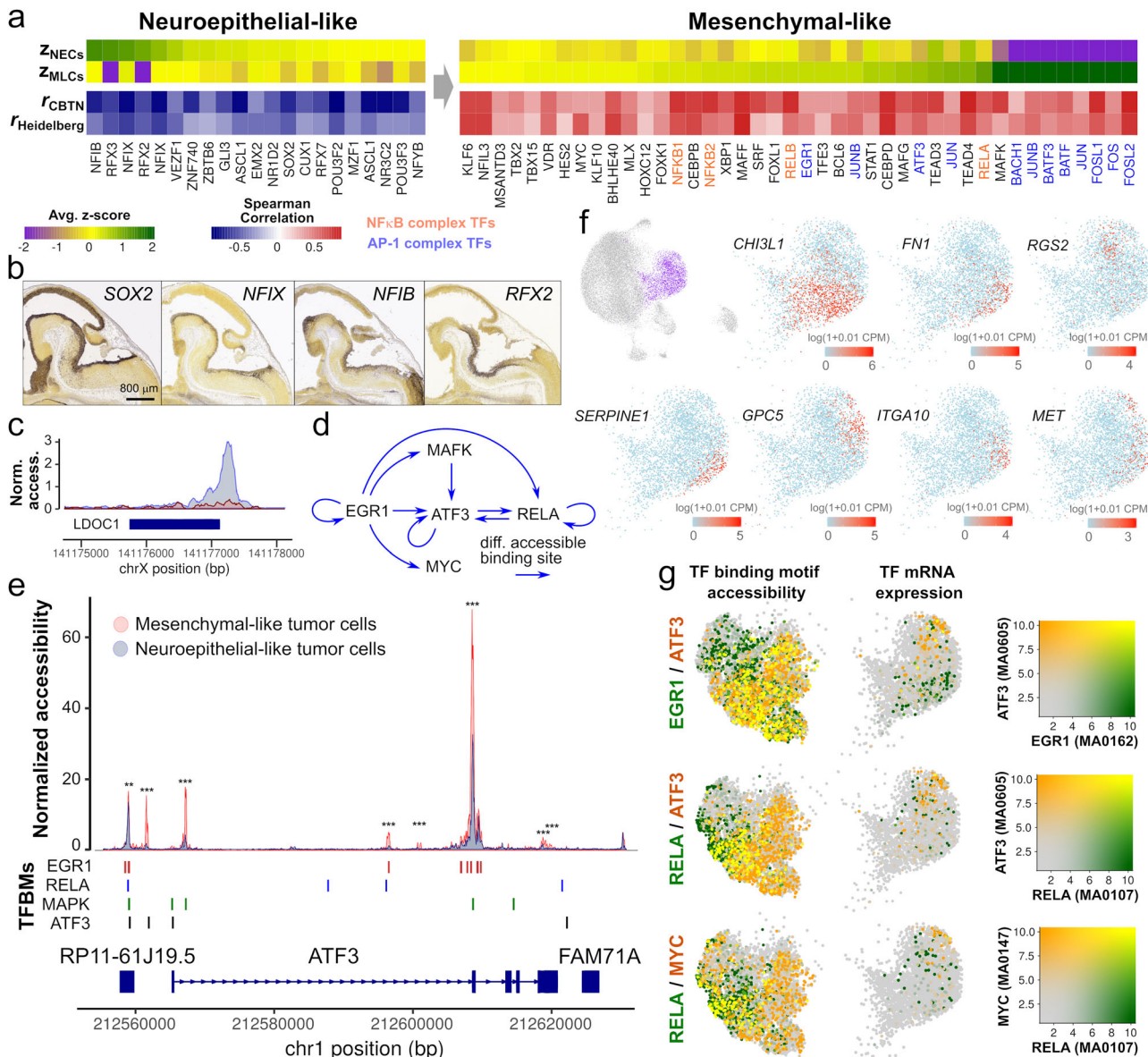

**Fig. 4 The transition of posterior fossa ependymoma tumor cells into a mesenchymal-like state involves the inhibition of neuroepithelial transcription factors and the activation of the NFκB and AP-1 complexes. a** Differentially expressed transcription factors in the neuroepithelial- or mesenchymal-like tumor cell populations that have differentially accessible binding motifs across the EMT (FDR < 0.1). The heatmap shows the transcription factor binding motif score in the neuroepithelial ($z_{NECs}$) and mesenchymal-like ($z_{MLCs}$) cell tumor cell populations according to the single-nucleus ATAC-seq data, and the Spearman's correlation coefficient between the inferred abundance of MLCs and the expression level of the transcription factor in the CBTN ($r_{CBTN}$) and Heidelberg ($r_{Heidelberg}$) bulk RNA-seq cohorts. Transcription factors that are part of the NFκB and AP-1 complexes are indicated in orange and blue, respectively. **b** Whole-mount RNA in situ hybridization of sagittal sections of E13.5 mouse embryos, showing the expression of neuroepithelial transcription factors in the neuroepithelium of the brainstem and isthmic organizer (Image credit: Allen Institute). **c** Normalized chromatin accessibility profile at the *LDOC1* locus for neuroepithelial-like and MLC populations, showing the inaccessibility of the promoter in MLCs. **d** Inferred regulatory network between some members of the MAF/BACH, NFκB, and AP-1 complexes. Arrows from one transcription factor into another indicate the presence of at least one differentially accessible binding motif of the first transcription factor in the gene locus of the second transcription factor (Fisher's exact test, FDR ≤ 0.05). This analysis indicates ATF3 integrates signals from the AP-1, ERK, MAF/BACH, and NFκB signaling pathways. **e** Normalized chromatin accessibility at the *ATF3* gene locus for neuroepithelial- and mesenchymal-like tumor cells, where the binding sites of EGR1, RELA, MAPK, and ATF3 are indicated. Transcription factor binding sites are differentially accessible between the neurepithelial- and mesenchymal-like cell populations. (Two-sided Fisher's exact test; *FDR ≤ 0.05, **FDR ≤ 0.01, ***FDR ≤ 0.001). Numeric *q* values are provided in the Source Data file. **f** The part of the UMAP representation corresponding to the MLC population is colored by the expression level of several genes with significantly heterogeneous expression within that cluster (Laplacian score, FDR < 0.01) based on a spectral graph method. **g** The part of the single-nucleus ATAC-seq and RNA-seq UMAPs that corresponds to the MLC population is colored by the transcription factor binding motif accessibility score and the gene expression level of RELA, ATF3, EGR1, and MYC transcription factors. In the figure, a substantial amount of heterogeneity is observed within the MLC population, and the relative patterns of expression and motif accessibility are consistent between the single-nucleus RNA-seq and ATAC-seq representations. MLCs mesenchymal-like tumor cells, NECs neuroepithelial-like tumor cells, TFBMs transcription factor binding motifs.

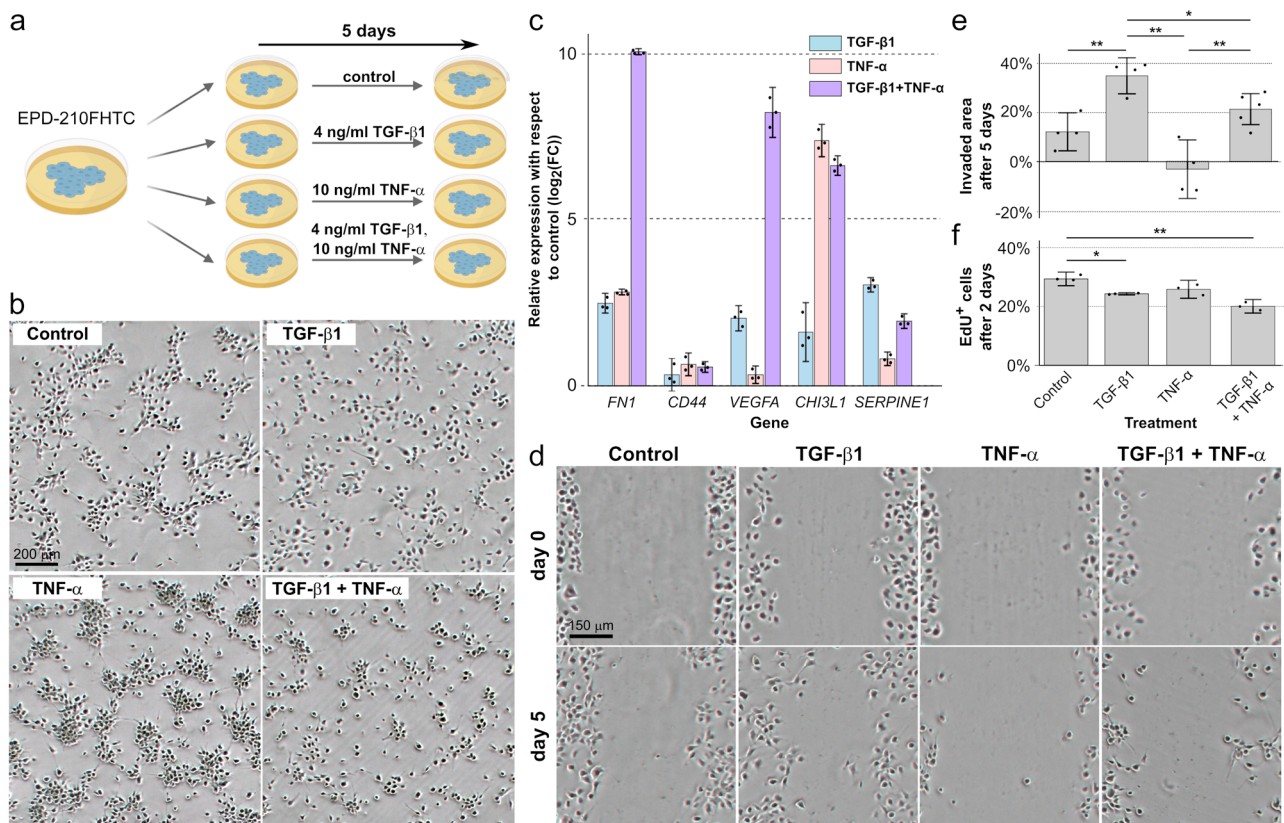

**Fig. 5 Pro-inflammatory cytokines induce the expression of mesenchymal-like genes and diverse cellular phenotypes in a patient-derived PFA cell model. a** Experimental design. The PFA primary cell line EPD-210FHTC was cultured in the presence or absence of TGF-β1 and/or TNF-α for 5 days. The experiment was performed in three biological replicates. **b** Image of the EPD-210FHTC cells after the 5-day culture, for each of the treatments. TGF-β1 is required for the cells to acquire a mesenchymal phenotype, whereas treatment with TNF-α potentiates the formation of 3D colonies. **c** Average change in the gene expression level of MLC-specific markers with respect to the no-treatment control after 5 days of treatment. Gene expression levels were profiled by RT-qPCR. Both TGF-β1 and TNF-α induce the upregulation of MLC markers, but the relative expression levels of *FN1* and *CHI3L1* strongly depend on the particular treatment. Error bars indicate 90% confidence intervals. **d** Cell migration assay. An example of the same gap at day 0 and day 5 is shown for each condition. Treatment with TGF-β1 leads to a substantial increase in the invasive potential of the cells. **e** Average fraction of the area invaded by the cells after 5 days in the cell migration assay across four biological replicates (two-sided *t*-test; *$p$ value $\leq 0.05$, **$p$ value $\leq 0.01$). Error bars indicate 90% confidence intervals. **f** Cell proliferation assay by EdU incorporation. The average proportion of EdU+ cells after 2 days of treatment is shown for each of the experimental conditions. The cell proportions in each condition were compared to the cell proportion in the no-treatment control across three biological replicates using a two-sided *t*-test. Treatment with TGF-β1 and/or TNF-α leads to a small reduction in cell proliferation (**$p$ value $\leq 0.01$, ***$p$ value $\leq 0.001$). Error bars indicate 90% confidence intervals. Source data and numeric $p$ values are provided in the Source Data file.

cell-cell junctions and grew in colonies, consistent with a neuroepithelial phenotype (Fig. 5b). However, treatment with TGF-β1 led to the disruption of cell-cell junctions, indicating a more mesenchymal morphological phenotype (Fig. 5b). RT-qPCR showed the upregulation of mesenchymal (*FN1*, *CHI3L1*, *CD44*, *VEGFA*, *SERPINE1*) marker genes in the presence of TGF-β1 (Fig. 5c), consistent with the induction of the EMT-like process by this cytokine. Treating the cells with TNF-α alone also induced the upregulation of these marker genes, including a 53x increase in *CHI3L1* expression with respect to the treatment with TGF-β1 alone (Fig. 5c, two-sided *t*-test between log$_2$(fold changes), $p$ value $< 10^{-4}$). However, cells grew in 3D colonies and had a non-mesenchymal morphological phenotype in the presence of TNF-α alone (Fig. 5b), indicating that TGFβ signaling is required to disrupt cell-cell junctions in this cell model. TNF-α in combination with TGF-β1 potentiated the mesenchymal gene expression signature induced by TGF-β1 (Fig. 5b) and led to a 193x increase in *FN1* (two-sided *t*-test $p$ value $< 10^{-3}$), a 31x increase in *CHI3L1* (two-sided *t*-test $p$ value $< 10^{-3}$), and 76x increase in *VEGFA* (two-sided *t*-test $p$ value $< 10^{-3}$) expression with respect to the treatment with TGF-β1 alone (Fig. 5c). To investigate if the

observed disruption of cell-cell junctions induced by TGF-β was associated with a more invasive phenotype, we evaluated the migratory capacity of the cells by means of a cell migration assay (Fig. 5d, e). Consistent with the mesenchymal morphological phenotype induced by this cytokine, the tumor cells significantly increased their invasive capacity upon treatment with TGF-β1 (two-sided *t*-test $p$ value $< 0.01$). To interrogate the proliferative potential of the cells under each condition, we labeled them with 5-Ethynyl-2′-deoxyuridine (EdU) and treated them with TGF-β1 and/or TNF-α for 2 days. Treatment with TGF-β1 and/or TNF-α led to a significant decrease in the rate of cell proliferation with respect to the non-treatment control (Fig. 5f, two-sided *t*-test between labeled cell proportions, $p$ value $< 0.01$), consistent with the lower proliferation rates of mesenchymal cells[75]. Taken together, these results confirm that pro-inflammatory cytokines can trigger a mesenchymal-like gene expression signature in posterior fossa ependymoma cells. However, the phenotype of cells expressing mesenchymal marker genes is diverse and depends on the specific cytokines that act on them. In particular, our study of this cell model suggests that TGF-β signaling is required for MLCs to acquire an invasive phenotype, which is

characterized by particularly high expression levels of *FN1*, *CHI3L1*, and *SERPINE1*.

## Discussion

Tumor cell states are the result of multiple concurrent molecular processes that are modulated by the genetic and epigenetic alterations of the cells. Here, we have studied the gene regulatory circuits that underlie cell-intrinsic differentiation programs and responses to the local microenvironment in pediatric posterior fossa ependymoma. Our study has particularly focused on the transition of tumor cells into a mesenchymal-like cell state. For that purpose, we have generated a single-nucleus chromatin accessibility and gene expression atlas of primary and metastatic posterior fossa tumors comprising ~40,000 cells. This atlas represents a unique resource for the study of the molecular programs associated with the cell ecosystem of this disease. Our results provide a more detailed and mechanistic understanding of the mesenchymal-like gene expression signature identified in previous single-cell RNA-seq studies of posterior fossa ependymoma[12,13]. The combined analysis of single-nucleus chromatin accessibility and gene expression data, as well as our experiments with a patient-derived cell model, provide evidence of the involvement of specific cytokines from the tumor microenvironment in the mesenchymal transformation of posterior fossa ependymoma, and identify the main transcription factors and regulatory elements involved in this process. These data show that the transition of tumor cells into a mesenchymal-like state is characterized by the inactivation of neuroepithelial transcription factors, such as SOX2 and members of the Nuclear Factor I and Regulatory Factor X families, and the activation of transcription factors from the NFκB, AP-1, MAF/BACH, MYC, YAP/TAZ, and sonic hedgehog signaling pathways. Given that the timeframe of RNA velocity trajectories in our analyses is determined by the rate of splicing and nuclear export[76], we expect the transition between neuroepithelial and mesenchymal-like cells to take place in hours. This time scale is consistent with our experiments on a patient-derived cell model and is comparable to the observed time scale for the mesenchymal transformation of glioblastoma pro-neural stem cells upon ionizing radiation[77].

Ependymoma mesenchymal-like cells are characterized by the expression of pro-inflammatory cytokines, hypoxia, angiogenesis, and glycolysis programs, which are reminiscent of the reactive gliosis programs that take place during brain injury and neuroinflammation. Although these programs are also expressed by mesenchymal cells during the EMT of epithelial cancers, our results also highlight important differences. Most notably, we find that the mesenchymal-like cell population, as defined by the expression of the mesenchymal-like gene signature, is a heterogeneous population of cells, and only a subset of it has the high-motility phenotype that is characteristic of bona fide mesenchymal cells. We therefore find that it may be more accurate to collectively refer to the mesenchymal-like cell population as "tumor-derived reactive glia". Our experiments with a patient-derived cell model suggest that TGF-β signaling is required to induce the high-motility phenotype in tumor-derived reactive glia, whereas other pro-inflammatory cytokines like TNF-α may potentiate this phenotype by crosstalk with the NFκB pathway. We have identified some of the molecular characteristics of the high-motility mesenchymal-like phenotype in our patient-derived cell model, most notably high gene expression levels of *FN1*, *CHI3L1*, and *SERPINE1* in relation to other tumor-derived reactive glia. However, more extensive studies using orthotopic mouse models are needed to fully characterize this population under physiological conditions and in relation to the molecular data from patients, and to clarify if the migratory phenotype observed in vitro is actively involved in tumor invasion and metastasis.

Our results also indicate a possible role of tumor-infiltrating microglia in the mesenchymal transformation and progression of ependymal tumors by acting as a source of TGF-β1 and other EMT-like inducing cytokines. The implication of the microenvironment in the mesenchymal transformation of glioma has been also noted in the context of glioblastoma[38,78]. As occurs with ependymoma, the mesenchymal gene expression signature of glioblastoma has been associated with abundant microglia infiltration, vascularization, and mesenchymal-like tumor stem cells[79,80]. The transition of pro-neural into mesenchymal glioblastoma stem cells is promoted by NFκB and YAP/TAZ signaling and can be triggered by hypoxia, reactive oxygen species, ionizing radiation, and genetic alterations such as the loss of the neurofibromin 1 (*NF1*) gene[17,66,77,81–83]. In the case of ependymal tumors, however, further work is needed to identify the specific triggers of mesenchymal transformation, which might include cellular stresses that are unique to ependymoma, such as inflammation and metabolic burden.

From a therapeutic perspective, our results suggest the importance of targeting both proliferative neuroepithelial-like tumor cells and tumor-derived reactive glia for prospective therapies to be effective. This can prove to be challenging due to the large differences that we observe in the gene regulatory circuits and pathways that control the maintenance and proliferation of these two stem cell populations. However, our results identify several transcription factors, such as MEIS1/2, that are active in both neuroepithelial-like tumor cells and tumor-derived reactive glia, and which seem to be specific to tumor cells. These are thus promising candidates to therapeutic targets, for which small-molecule inhibitors already exist[47], and which deserve to be further studied. More broadly, we expect that the combination of high-throughput molecular profiling techniques, such as those used in this study, with perturbation experiments using orthotopic mouse models will enable the development of new therapeutic approaches for pediatric ependymoma.

## Methods

**Ethical approval**. All procedures in this study were performed according to the institutional regulations of the University of Pennsylvania and the Children's Hospital of Philadelphia. The specimens and data were provided by the CBTN in a deidentified form according to the U.S. Department of Health and Human Services regulations and were not considered as Human Subjects Research by the Institutional Review Board of the University of Pennsylvania.

**Tumor samples**. De-identified flash-frozen tumor specimens were provided by the CBTN biorepository (Approved Biospecimen Project #29). The anatomic location of the tumors and their diagnosis were obtained from the deidentified surgical, radiology, and pathology reports. We considered samples for which the age at the time of diagnosis was <16 years old, their location was in the posterior fossa (for primary tumors and recurrences) or were derived from a primary tumor located in the posterior fossa (for metastases), and their RNA was well preserved according to bioanalyzer (see "Single-nucleus RNA-seq library preparation and sequencing"). The molecular identity of the tumors was confirmed based on the expression of gene markers in bulk RNA-seq data (see "Bulk RNA-seq data processing"). For specimens with no available bulk RNA-seq data (I2 and M8), the molecular identity was assessed by quantitative reverse transcription PCR. cDNA synthesis was performed using Maxima H Minus Reverse Transcriptase (Thermo Scientific, cat. # EP0753) according to the manufacturer's protocol, with Oligo(dT)18 (Fisher Scientific, cat. # SO131). cDNA concentration was measured with a Qubit 3 Fluorometer (Life Technologies). We used 10 ng of cDNA for each RT-qPCR reaction and KiCqStart SYBR Green primer pairs for *GAPDH*, *L1CAM*, *APLNR*, *IFT46*, *CXorf67*, and *MECOM* (Sigma-Aldrich). SYBR FAST Universal qPCR Master Mix with low Rox (Roche Sequencing, cat. # KK4602) was used and the reaction was performed with a QuantStudio 7 Flex (Applied Biosystems). Four technical replicates of each sample-primer combination were performed and used to determine the standard error.

**Single-nucleus RNA-seq library preparation and sequencing**. Approximately 8 mm³ of tissue was cut from each tumor. Samples were incubated at room

temperature in 1 ml homogenization buffer for 3 min, then dissociated with 13 strokes of a tight pestle in a 2 ml glass homogenizer. Samples were then filtered with a 40 μm mesh filter followed by a 30 μm mesh filter. After brief tabletop centrifugation, the nuclei pellet was resuspended in 1 ml PBA-BSA 0.01% and counted. Nuclei were defined by shape, size, and general appearance, and diluted to 100 nuclei/μl. Single-nucleus RNA-seq library preparation was performed as previously described[26] using a Drop-seq microfluidic system. The quality of the cDNA libraries was evaluated with a bioanalyzer and quantified using a KAPA Library Preparation Kit (Roche Sequencing, cat. # KK4824). Libraries were sequenced using an Illumina NextSeq 500 on high output mode with 20 bp (read 1) and 60 bp (read 2) paired end reads.

**Single-nucleus ATAC-seq library preparation and sequencing.** Nuclei were isolated from flash-frozen tumors following the 10x Genomics demonstrated protocol for Mouse Brain Tissue, with the following modifications: the first incubation in the lysis buffer was decreased to 45 s, and the second incubation, after pipet mixing, was 90 s. A 40 μm nylon cell strainer and a 30 μm MACS Smart-Strainer (Miltenyi Biotec, cat. #130-098-458) were used in place of the suggested 70 μm and 40 μm tip strainers. Single-nulcei ATAC-seq cDNA libraries were prepared using the Chromium Single-Cell Platform (10x Genomics) by the Center for Applied Genomics of the Children's Hospital of Philadelphia and sequenced on an Illumina NovaSeq 6000 sequencer using two 100-cycle SP flow cells.

**Cell culture experiments.** The EPD-210FHTC cell line was acquired from the Brain Tumor Resource Lab at Fred Hutchinson. This early passage cell line has been recently authenticated by the Brain Tumor Resource Lab using whole-genome methylation and gene expression profiling[74]. We confirmed the expression of posterior fossa ependymoma stem cell markers (*SOX2, VIM, TKTL1, GRIA4*) upon receiving the cell line. The cell line was tested negative for mycoplasma contamination using a PCR based method. Growth conditions were followed as suggested by the Brain Tumor Resource Lab. Cell culture media had the following components: NeuroCult NS-A Proliferation Kit (Human) (Stemcell Technologies, cat. #L05751) supplemented with 20 ng/ml human FGF (Peprotech, cat. #100-18B), 20 ng/ml murine EGF (Peprotech, cat. #315-09), and 1% penicillin/streptomycin (Gibco, cat. #15140122). Plates were coated with laminin (Sigma-Aldrich, cat. #L2020) (10 μg/ml) for 6 to 24 h at 37 °C before use. Cells were cultured at 37 °C, 5% $CO_2$. For TGF-β1 and TNF-α treatments, cells were treated with 4 ng/ml TGF-β1 (Sigma-Aldrich, cat. #H8541-5UG), 10 ng/ml TNF-α (Sigma-Aldrich, cat. # H8916-10UG), or both, for 5 days before RNA extraction. Media was replenished every 48 h. Cells were plated at the same density and given 24 h after plating before receiving treatment. After 5 days, cells were imaged and RNA extraction was carried out using the RNAqueous 4PCR kit (Life Technologies, cat. #AM1914). RNA was immediately stored at −80 °C until cDNA synthesis. cDNA was synthesized using Maxima H Minus Reverse Transcriptase (Thermo Scientific, cat. #EP0753) following the manufacturer's instructions. KiCqStart SYBR Green primer pairs for *ACTB, FN1, VEGFA, CHI3L1, SERPINE1*, and *CD44* (Sigma-Aldrich) were used for RT-qPCR. All experiments were done in at least three biological replicates.

**Proliferation assay.** Cells were plated on culture slides (Falcon, cat. #354114) treated with laminin (10 μg/ml) 24 h before receiving treatment with TGF-β1, TNF-α, or both, as described in the paragraph "Cell culture experiments", as well as EdU (Click-It Edu Kit for Imaging, Invitrogen, cat. #C10338) at a concentration of 10 μM. After 48 h, slides were fixed and permeabilized, followed by staining with AlexaFluor 555 provided by the Click-It kit per the manufacturer's protocol. Images were taken using a Leica TCS SP8 Multiphoton Confocal Microscope. The proportion of EdU+ nuclei was quantified using ImageJ (version 2.35). We despeckled the images and used an adaptive local threshold based on moments to delineate signal from background pixel intensity. We then removed outlier pixels based on a local radius, used the binary open function of ImageJ to remove background signal originating from the cell soma, and eliminated the remaining noise by another round of despeckling and removing outlier signal. We used the watershed algorithm to segment nuclei based on the nuclear stain and filtered out regions with an area smaller than 15 or larger than 600 μm. Segmented nuclei were added to the ImageJ ROI manager and used to count the proportion of EdU+ nuclei. To identify EdU signal from background, we despeckled the images from the corresponding channel and used an adaptive local threshold based on moments. We used a *t*-test to determine whether the mean proportion of EdU+ nuclei within each of the three treatment conditions was equal to the control condition. The assay was performed in three biological replicates.

**Cell migration assay.** Cells were plated on laminin-coated 6-well plates and allowed to grow for 24 h. A P200 pipet tip was used to then generate a wound on the cell monolayer. Cell media was immediately changed to fresh media containing appropriate treatment (TGF-β1 at 4 ng/ml, TNF-α at 10 ng/ml, or both cytokines). Wounds were imaged once every 24 h for 5 days. The assay was performed in four biological replicates.

**Bulk RNA-seq data processing.** De-identified raw RNA-seq data of 44 out of 46 pediatric ependymal tumors located in the posterior fossa were provided by the CBTN biorepository[84] (CBTN Approved Data Project 19). Gene expression was quantified using Kallisto (version 0.45.1)[85] and aligned to the human reference genome GRCh38. To assign each of the 44 tumors a molecular group, we developed a classification method based on gene set enrichment analysis (GSEA) (https://zenodo.org/badge/latestdoi/264224369). This method determines whether a gene signature $s$ is significantly enriched among a ranked list of genes $L$ ordered by expression. Gene signatures $s$ were constructed from an independent cohort of ependymal tumors profiled for expression by microarrays[5]. For each of the eight molecular groups in this cohort (supratentorial sub-ependymoma, supratentorial ependymoma with YAP1-fusion, supratentorial ependymoma with RELA-fusion, posterior fossa sub-ependymoma, posterior fossa ependymoma group A, posterior fossa ependymoma group B, spinal myxopapillary ependymoma, and spinal anaplastic ependymoma), we used limma's linear model (version 3.34.1)[86] to identify the top $n = 50$ differentially expressed genes exclusively upregulated in each group ($p$ value < 0.01 and log2 fold change >2). Additionally, we utilized the gene signatures for PFA-1 and PFA-2 tumors from the bulk transcriptomic data of ref. [8], consisting of the top $n = 58$ upregulated genes exclusive to each signature. To determine the molecular subgroup of each tumor in the CBTN cohort, we performed GSEA for each signature on the query bulk RNA-seq data. For that purpose, we calculated a running-sum statistic for each signature $s$ by walking down the complete list $L$ of genes in the query dataset in decreasing order of expression. For gene $j$ in $L$, the statistic increases by $\frac{1}{n}$ if $L_j \in s$ or decreases by $\frac{1}{m-n}$ if $L_j \notin s$, where $m$ is the total number of genes in $L$. By randomizing the genes in $s$, we used a permutation test to estimate the statistical significance of each score. If the $p$ value of all signatures was above 0.35, no molecular group was assigned. Due to apparent limited power of the PFB gene expression signature, tumors were classified as PFB if their PFB $p$ value was less than 0.34. The training error of our classifier was 1.4% (3/209). To further benchmark the performance of our classifier, we ran it on an extended cohort of 94 pediatric ependymal tumors of mixed anatomical origin from the CBTN biorepository for which bulk RNA-seq data were available. The classifier correctly predicted the anatomic location of the tumors in 92.6% (87/94) of the cases.

**Single-nucleus RNA-seq data processing.** The Drop-seq computational pipeline[87] was used to map to the human reference genome (GRCh38) via STAR alignment (version 2.6.1a)[88], correct the cellular and molecular barcodes for sequencing errors, and build a count matrix. To account for poly-adenylated pre-mRNAs in the nucleus, we included reads that aligned to intronic regions of the genome (locus_function_list = intronic). We removed low-quality cells, debris, and empty droplets based on the inflexion point in a plot of the number of UMIs in each droplet ranked by decreasing order of magnitude. Additionally, we plotted the number of UMIs (in log scale) against the proportion of mitochondrial genes for each droplet. For most samples, the distribution of droplets in this scatter plot was bimodal. We used a linear cut to segregate cells (high number of UMIs and low percentage of mitochondrial genes) from debris (high percentage of mitochondrial genes and low number of UMIs) in this plot. Gene expression was log-normalized by library size in each cell and the most variable genes were selected for PCA using Seurat (version 3.2.2)[89]. We then used Harmony (version 0.1.0)[27] with default parameters to build a consolidated representation of the nine samples based on the top 5000 most variable genes and 30 principal components. We clustered the data using Louvain community detection based on the top 30 dimensions of the consolidated space and used Uniform Manifold Approximation and Projection (UMAP)[90] to construct a two-dimensional visualization.

**RNA velocity field.** We used the Velocyto command line tool[76] to select for spliced and unspliced reads in each cell and generate loom files. We then used the package scVelo[28] to estimate the RNA velocity field using dynamical modeling based on the top 15 principal components, 30 nearest-neighbors, and the top 2000 most variable genes with at least five counts in total. To apply this model to single-nucleus RNA-seq data we reinterpreted the RNA degradation rate of the scVelo model as the nuclear sport rate. Using the same set of parameters that we used for clustering and UMAP visualization (cf. "Single-nucleus RNA-seq data processing") led to a comparable RNA velocity map, except for an overall directionality in the IPC population.

**Differential gene expression analysis.** For each sample, we used edgeR's general linear model (version 3.20.1)[91] to identify differentially expressed genes between each cell population and the other cells in the sample. Cell population assignments were based on the combined analyses of all samples. We only considered genes expressed in at least 5% of the cells of the cluster and samples where at least 3% of the cells belonged to the cluster of interest. We then used Fisher's $\chi^2$ statistic to combine the $p$ value of each gene across samples for genes expressed in at least two samples. We followed the same approach to perform differential expression between the mesenchymal- and neuroepithelial-like tumor cell populations. We used the package CellChat[35] to identify differentially expressed genes based on an un-truncated mean, assign genes to ligand-receptor/co-receptor pairs, and categorize them into signaling pathways. We used the R package RayleighSelection[67] to

compute the Laplacian score on the log-normalized expression matrix and identify genes with a significant gradient of expression within the MLC population. We used Pearson's correlation distance as metric and took the radius parameter ε of RayleighSelection to be the median pairwise distance among cells. We only considered genes expressed in 4–50% of the cells in this analysis. To reduce the runtime, we used a random sample of 1000 cells and verified the consistency of the results across different sets of randomly sampled cells.

**Gene-set enrichment analysis**. We used the R packages msigdbr[92,93] and fgsea[94] to download the hallmark, HINATA_NFKB_TARGETS_FIBROBLAST_UP, and HINATA_NFKB_TARGETS_KERATINOCYTE_UP gene modules from the MSigDB database. We defined a NFKB gene signature by taking the intersection between the two HINATA gene modules. We computed GSEA normalized enrichment scores and $p$ values of these gene modules and the mesenchymal gene module from ref. [42] in each dataset. When applied to single-nucleus RNA-seq data, we first aggregated cells into a synthetic bulk RNA-seq dataset. GOE analyses were performed with g:Profiler[95], using the ordered list of differentially expressed genes with FDR ≤ 0.01 in the cell population of interest as the query and adjusting the $p$ values for multiple hypotheses testing using the recommended g:SCS method. Upper bounds on the FDR were estimated using the Benjamini-Hochberg procedure.

**Deconvolution of bulk RNA-seq data**. We used CIBERSORTx[44] to infer cell type abundances from bulk gene expression data. To do this, we used CIBERSORTx to build a gene expression signature matrix of the cell populations identified in the single-nucleus RNA-seq data (with min_expression = 0) and infer cell type abundances in the TPM-normalized bulk transcriptomic data using the S-mode batch correction. We subsampled large populations to 1000 cells and checked that different samples did not significantly affect the results of the deconvolution.

**Single-nucleus ATAC-seq data processing**. We used Cell Ranger ATAC (version 2.0) to map the FASTQ files to the human reference genome (GRCh38) and demultiplex by their cell barcode. We used Signac[96] to call and quantify peaks. We selected high-quality cells based on the number (>1500 and <35,000) and percentage (>20%) of fragments in peaks, the ratio of mono-nucleosomal to nucleosome-free fragments (<3), and the TSS enrichment score (>2). In addition, we filtered out cells based on the percentage of mitochondrial fragments (>10%) and the percentage of fragments overlapping targeted sites (<20% or <25%, depending on the sample). We removed peaks overlapping the ENCODE blacklisted regions.

To integrate the single-nucleus ATAC-seq. data into a consolidated representation, we created a union peak set by merging intersecting peaks across samples using the reduce function from the GenomicRanges package (version 1.38.0). We removed peaks whose width was more than 3 standard deviations greater than the mean peak length and concatenated the union peak sets from each sample. For normalization and dimensionality reduction, we used latent semantic indexing[97]. To do this, we used peaks that were open in at least 90% of the cells for term frequency-inverse document frequency normalization. The top 30 dimensions were integrated with Harmony (with parameters lambda = 0.5 and theta = 1). After removing the first dimension, the top 30 harmony dimensions were clustered using Louvain community detection and visualized with UMAP. To infer the gene expression associated with each cell, we built a gene activity score matrix by summing fragments that overlapped with a gene body or promoter (defined as the 2 kb region upstream of the TSS). We merged two clusters that have high gene activity scores of VEGFA, CD44, and MET into a single-cell population of MLCs. The vascular cell population was sub-clustered to separate the clusters of endothelial and mural cells.

**Analysis of large-scale copy number alterations**. We used Copy-scAT[45] to infer large-scale copy number alterations from the single-nucleus ATAC-seq data. We used Copy-scAT's semi-supervised approach to identify normal and neoplastic cells in each sample. Copy number alterations were inferred in each cell using the identifyCNVClusters function and the default parameters suggested in the documentation. We only considered alterations present in at least 50% of the cells in a cluster.

**Differentially accessible of cis-regulatory elements**. After removing peaks expressed in less than 30% of cells in a cluster, we used a Fisher exact test in each sample to identify differentially accessible peaks between each cell population and all the other cells in the sample. Cell population assignments were done based on the combined analyses of all the samples. In the analysis of enhancers, we only kept peaks that overlapped regions in the GeneHancer database[51]. We used Fisher's $\chi^2$ statistic to combine $p$ values associated with peaks across samples for peaks that were accessible in at least two samples (except for mural cells which were only composed on one sample). Samples with less than 4% of the cells belonging to the population of interest were not included in the combination.

**Motif enrichment analysis**. We used chromVAR[46] to infer the bias-corrected transcription factor deviation score of each motif from the JASPAR2020 database[98] in the union peak set. For each sample, we used a Wilcoxon rank-sum test to identify transcription factor deviation scores upregulated in each population compared to the rest of the cells in a sample. Cell population assignments were done based on the combined analyses of all the samples. We then used Fisher's $\chi^2$ statistic to combine the $p$ values of each transcription factor across samples. Samples with less than 4% of the cells belonging to the population of interest were not included in the combination. For visualization, we removed the top and bottom 10% positive scores for each cell, as described in ref. [96]. We computed the Laplacian score of the transcription factor deviation score using a random sample of 1000 cells from the MLC population to identify transcription factors that have a substantial amount of variability in their activity within this population.

**Regulatory network between transcription factors with differentially accessible binding motifs across the EMT-like process**. To reconstruct the regulatory network between members of the MAF/BACH, NFκB, and AP-1 complexes with differentially accessible binding motifs across the EMT-like process (Fig. 4d), we downloaded the location of the binding sites of these transcription factors from the JASPAR2020 database[98] and overlaid the location of differentially accessible peaks between neuroepithelial- and MLC populations (as described in "Differentially accessible of cis-regulatory elements"). A network was then built by taking the transcription as nodes and adding a directed edge from one transcription factor into another if one or more significant ($p$ value < 0.05) differentially accessible binding sites of the first transcription factor were present in the gene locus (±10 kbp upstream and downstream) of the second transcription factor.

**Reporting summary**. Further information on research design is available in the Nature Research Reporting Summary linked to this article.

## Data availability
The raw and processed single-nuclei RNA and ATAC-seq data generated in this study have been deposited in the Short Read Archive (SRA)/Gene Expression Omnibus (GEO) databases with accession number GSE206580. The publicly available bulk RNA sequencing data[25] used in this study are available from the Kids First Data Resource Portal (https://portal.kidsfirstdrc.org, project PBTA-CBTN). The publicly available gene expression data of the Heidelberg ependymoma cohort[5] used in this study are available from the GEO database with accession code GSE64415. The JASPAR2020 database[98] used in this study is publicly available at https://jaspar.genereg.net/downloads/. The publicly available hallmark gene signatures[92] used in this study are available from the MSigDB database (https://www.gsea-msigdb.org/gsea/msigdb/collections.jsp#H). The mouse embryo RNA in situ hybridization data used in this study are available from the Allen Developing Mouse Brain Atlas (https://developingmouse.brain-map.org/). The GRCh38 human reference genome is available from Ensembl (http://ftp.ensembl.org/pub/release-106/fasta/homo_sapiens/dna/). The remaining data are available within the Article, Supplementary Information, or Source Data file. Source data are provided with this paper.

## Code availability
The code used for the classification of ependymal tumors based on their gene expression is available at zenodo and the corresponding DOI is as follows: https://doi.org/10.5281/zenodo.6607426[99].

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

## Acknowledgements

The authors are grateful to the CBTN for providing data and tissue specimens for conducting this study. They also thank Jim Olson and Emily Girar for assistance with the EPD-210FHTC cell model, Peng Hu, Qi Qiu, and Hao Wu for assistance with the implementation of sNuc-Drop-seq, Jen Crainic, Terence Gade, Yugong Ho, Julie I-Ju Leu, Stephen Liebhaber, Richard Phillips, Jennifer Phillips-Cremins, and Celeste Simon for assistance with various experimental aspects, Doug Epstein for relevant scientific discussions, Rushabh Mehta for collaboration on related topics, the Next-Generation Sequencing Core of the University of Pennsylvania for assistance with cDNA library sequencing, and the Center for Applied Genomics of the Children's Hospital of Philadelphia for assistance with single-nucleus ATAC-seq library preparation and sequencing. This work has been supported by the ASPIRE Award 20-044-ASP (P.G.C.) from The Mark Foundation for Cancer Research, and an Advisory Council Research Award (P.G.C.) from the CBTN. The work of R.G.A. was partially supported by the NIH NHGRI T32 Training Grant T32HG00046. The work of J. M. was supported by the NIH NHGRI R25 Education Program R25HG010323.

## Author contributions

R.G.A. carried out the computational analyses. E.C.T. performed the experiments. A.N.A. and J.M. assisted with the computational analyses. M.S. and M.P.N. provided histopathology expertise. R.G.A., E.C.T., and P.G.C. wrote the manuscript. P.G.C. conceived and supervised the study.

## Competing interests

The authors declare no competing interests.
