## [Peer Review File · Nature Communications]

Pro-Inflammatory Cytokines Mediate the Epithelial-to-Mesenchymal-Like Transition of Pediatric Posterior Fossa EpendymomaReviewers' Comments:

Reviewer #1:

Remarks to the Author:

Aubin, et al. reported the mesenchymal transition of ependymoma was induced by the pro-inflammatory cytokines in the tumor microenvironment, by means of single-cell chromatin accessibility analysis and gene expression profiling. They identified NF κ B, activator protein-1, and MYC as important mediators of this transition. This is a very interesting paper which discriminates tumor cells from the cells of microenvironment- composing cells such as astrocytes and microglia as the cytokine producing cells for the first time in this field.

For the benefit of the reader, some points need clarifying and certain statements require further justification. These are given below.

- 1) The authors insist that the source of TGF β are the mesenchymal-like tumor cells (autocrine) and astroblast (paracrine). If so, what is the first event in the EMT of ependymoma? Although the main cause of EMT is hypoxia, ependymoma typically has sparse necrotic areas in the pathological examination. Also, astroblast cannot usually release a large amount of TGF β .
- 2) As the authors described that the abundance of mesenchymal-like tumor cells was strongly correlated with the abundance of microglia, it is natural to interpret the infiltrated microglia is the main source of TGF β . Microglia is mesodermal origin and is high producer of TGF β .
- 3) Please explain the identification of astroblast in the UMAP of the single-nuclei RNA-seq analyses appeared in Figure 1. The definition of astroblast is not easy. Are these cells positive for GFAP and S-100 protein, or other molecular markers are definitely positive? The origin of astroblast may be the mesenchymal-like tumor cells in this study. There are a subtype of tanyctic ependymoma which has features intermediate between astrocytes and ependymal cells. More detailed pathological diagnoses of the original 41 tumors are necessary.
- 4) The comparison between the original and metastatic tumors is tricky, because the metastatic tumors usually experienced radiotherapy. Have all the tumors experienced radiotherapy? This point should be described in the Results section and mentioned in the Discussion section.

Reviewer #2:

Remarks to the Author:

The authors study the gene regulatory circuits in pediatric posterior fossa ependymoma to better understand how tumor cells transition to a mesenchymal-like state. They provide a single-nuclei chromatin accessibility and gene expression atlas of primary and metastatic posterior fossa ependymoma tumors through single RNA seq and ATAC seq data. The authors find the transition of tumor cells into a mesenchymal-like state is characterized by the inactivation of neuroepithelial transcription factors, such as SOX2 and members of the Nuclear Factor I and Regulatory Factor X families, and the activation of transcription factors from the NF κ B, AP-1, MAF/BACH, MYC, Hippo, and sonic hedgehog signaling pathways. The mesenchymal-like cell population is characterized by the expression of pro-inflammatory cytokines, hypoxia, angiogenesis, and glycolysis programs.

Overall, the manuscript is novel with results that are noteworthy with strong methodology. The results of the study are significant to the field of neuro-oncology and contributes to the existing literature. One criticism of the approach is only 1 patient tumor was included that was recurrent.

1. The MLC population is characterized by the mesenchymal-like gene signature; however, the authors find this population quite heterogeneous. What percentage of tumor cells are comprised by these tumor cells?
2. What WHO grade were the tumor studied in both primary, recurrent, and metastatic tumors?
3. The authors refer to the MLCs as tumor-derived reactive glia, how resistant are these cells to standard therapies such as radiation therapy? Do these cells have gene expression that suggests they

have better DNA repair mechanisms to resist radiation therapy treatment? Do they have expression of genes suggestive of chemotherapy resistance (e.g., ABC transporters, miRNA expression), are resistant to apoptosis, and have hypoxic stability?

4. Do the authors believe the tumor-derived reactive glia may represent a type of tumor stem cell that may propagate tumors and make them more aggressive clinically?

5. A higher motility phenotype is suggested by the MLC phenotype of tumors in the manuscript, were these cells more abundant in the tumor population studied from the metastatic and recurrent tumors?

6. The authors show that the transition of posterior fossa ependymoma tumor cells into a mesenchymal like state involves the inhibition of neuroepithelial transcription factors and the activation of the NFkB and AP-1 complexes. What do the authors propose is the switch that causes this transition of PFA tumor cells?

7. Did the authors find the microglia cell populations to be more abundant within each tumor type studied (primary, recurrent, and metastatic)? Are the tumor associated microglia more abundant within the recurrent and metastatic tumors? Are they responsible for the EMT switch in tumors with their association with MLCs? Are the microglia recruited from the circulation in PFA tumors?

8. Could the microglia account for a more invasive phenotype associated with the MLCs?

Reviewer #3:

Remarks to the Author:

In the manuscript "Pro-Inflammatory Cytokines Mediate the Epithelial-to-Mesenchymal-Like Transition of Pediatric Posterior Fossa Ependymoma" Rachael G. Aubin et al. performed single cell chromatin accessibility and transcriptomic profiling of primary ependymomas in the posterior fossa and metastasis in spine and cortex to reveal transcription factors and enhancers associated with differentiation of tumor cells to amongst other mesenchymal-like cells (MLC). Using ligand-receptor pair analysis from snRNA-seq they infer proinflammatory signaling between microglia and MLC and show in a patient derived cell model that cells with mesenchymal gene expression signature show diverse phenotypes depending on stimulation with TGFβ and/or TNFα.

I have several major concerns regarding the study, which are listed below:

1) Please specify how sample were selected to be included in snRNA-seq and snATAC-seq analysis? The number of biosamples for single cell analysis is comparably low and the samples are spread across many sub-types (PFA-1, PFA-2, PFB) and stages (initial, progressive, and distal metastasis). Please specify what PFB and PFA* correspond to and how they relate to PFA-1 and 2. Validation experiments performed in cell model from a cell line derived from a recurrent PFA ependymoma. Given the tumor heterogeneity and different trajectories cells might take I was wondering what cell types/stage does EPD-210FHTC correspond to or is it a mixture?

2) Sample M9 (and same is true to lesser extent for M7) which according to snRNA-seq data consists of 60% of MLC (Fig. 1E) and thus has the highest fraction of MLCs, but is listed somewhere in the middle in the analysis of inferred fraction of MLC from bulk RNA-seq (2C) and does not support the pathways hypoxia, glycolysis etc which the authors associate with the MLC population. Indeed this sample seems depleted for TNFα signaling through NFkB. This puts into question the conclusions related to the gene programs active in MLCs. Do the other samples to the left in 2C have indeed a higher fraction of MLCs? Based on Figure 1E this seems at least unlikely since all I_ samples have <5% of MLCs. How do the authors reconcile this discrepancy?

3) RNA velocity analysis is based on comparing spliced and unspliced RNA molecules and it seems there is no consensus if the assumptions and models are applicable to profiling of single nuclei. Thus, it is unclear what to make out of Fig 1F and if the indicated relationships and trajectories between cell types can be concluded from the data itself even if in agreement with the literature.

4) From the presented data it is difficult to follow the proposed EMT-like process because the MLC cluster consists of >90% cells from the 2 metastasis samples. Thus, it is possible that cells changed their phenotype and expression profile in response to the microenvironment or prior to migration or these are cells of different origin rather than transitioned. Here it could be interesting to look within one patient into both tumor and metastasis or compare MLC within a tumor sample to the corresponding other clusters while excluding MLCs from metastases samples. To illustrate that these are derived from tumor cells the ATAC data with inferred copy number variations could be helpful, since MLCs (if transitioned) would have the same CNVs as the origin cell. According to 3D both tumors with chrom. aberrations had no MLC. A snATAC-seq dataset of samples with MLC and chrom aberrations might help to address this.

5) Please explain why TFs differentially expressed between NE and MLC were used to infer EMT regulators? Based on the trajectory displayed there is no direct relationship between NE and MLC. Why not compare to undiff. cells or any of the other cell types?

6) Fig. 3A how were NPC separated from undifferentiated cells by the indicated dotted line? Fig 3B there is a huge increase in the fraction of MLCs in I2 and I5 > 5% in RNA to ~40% in the ATAC dataset. Was a very different region of the tumor used or is this linked to lower resolution and thus separation of clusters in ATAC? Since MLCs are a major focus of this study these drastic differences in composition should be addressed, e.g. by 2 technical repeats from a tumor.

7) Please provide metadata for each single cell experiments such as how many cells per data set, genes and transcripts/nucleus for RNA, fragments/nucleus and a measure for signal-to-noise ratio for snATAC-seq such as TSS enrichment.

Minor Point:

8) Please show marker genes for undifferentiated tumor cells as well (1B)? How were tumor from non-tumor cells distinguished in snRNA-seq data?

9) Fig. 1E and Fig. 3B stack plot to display contributions of samples to different clusters would allow for easier interpretation.

Reviewer #4:

Remarks to the Author:

Aubin et al. characterize the neuroepithelial-like tumor cells and mesenchymal-like tumor cells at the single-cell level using RNA and ATAC modalities in their manuscript titled "Pro-Inflammatory Cytokines Mediate the Epithelial-to-Mesenchymal-Like Transition of Pediatric Posterior Fossa Ependymoma". The authors investigate gene-regulatory mechanisms that lead to EMT-like phenotype in ependymoma and seek to understand the relationship of this process with inflammation. While the work has potential to be impactful, the manuscript itself has several shortcomings and need to be addressed.

Generally, a lot of the examples in the manuscript (for example choice of genes to investigate) seem to be chosen without providing any data-driven motivation. This also extends to computational methods where a lot of details are missing.

More specific comments:

1. In the introduction, the authors suggest that prior studies (Gillen et al. 2020, Gojo et al. 2020, Griesinger et al. 2015, Griesinger et al. 2017) have already described EMT-like process that involves NFkB activation and hypoxia that aids growth and propagation of the tumors. How the results of the authors differentiate from the prior studies is not revisited at the conclusion of the manuscript. We encourage authors to clearly differentiate their work.

2. How were the 10 sub-populations identified in Figure 1A annotated? Is the heatmap in Fig 1B showing the top differentially expressed genes? And was this heatmap used to annotate the clusters? Or was it the other way around? We suggest the authors to clearly state how the obtained clusters were annotated as specific cell types? It is also not clear how the undifferentiated cells were identified.

3. (A) The authors use scVelo to construct RNA velocity trajectories. The authors are requested to comment on the time frame of tumor progression and the time frame on which the assumptions made in scVelo operate (that of unspliced/spliced turnover); and to interpret scVelo results accordingly when applied to transition that operates at a different time scale. (B) The authors use a different set of parameters to run scVelo as opposed to UMAP/Clustering. We request the authors to motivate this.

4. At the end of the first section in "Results" the authors claim "numerous genes coding for components of the WNT signaling pathway were upregulated in non-mesenchymal" and present a few example genes. Why were only these specific genes chosen? The authors are requested to perform GSEA and show that WNT pathway is enriched here. Or at least show that genes of WNT pathway have a high expression using a proper gene signature as opposed to a set of few genes.

5. The authors seem to directly apply batch correction method. However, it would be more informative if they could provide reasoning for this. Perhaps, show visual of the data before and after the correction and argue why batch correction needs to be done. Tumors have biological variance between patients, how do the authors know that the variance they see (if they do) between batches is not biological? Perhaps the authors can highlight non-tumor cells and use it as a guide to show effect of batch correction (ideally, non-tumor cells like immune cells should be similar in phenotype across patients). Also, the authors use Harmony to correct batch effect, but Harmony does not correct for gene expression. How do the authors ensure their downstream calculations are not impacted by this? Is this done by computing differential expression in each sample separately? We request the authors to be explicit about this.

The comment on batch effect correction also applies for ATAC-seq data.

6. In the analysis using CellChat, how are the pathways identified? Is there any quantification to rank them? And how are the genes discussed chosen for? Also, CCL seems to be higher from microglia -> MLCs. Is this consistent with the authors' claims? This section generally needs more details and the conclusion made is not clear in how it follows from the data shown.

7. Details on how the genes (HIF3A and HIF1A) are selected for in the section: "Mesenchymal-like tumor cells are associated with abundant vascularization and microglia infiltration, and have elevated expression of NFkB target genes" are needed.

8. At the end of the section "Mesenchymal-like tumor cells are associated with abundant vascularization and microglia infiltration, and have elevated expression of NFkB target genes", the authors are requested to show the switch in expression at a signature level as opposed to select genes. Also, the authors are requested to comment on why the switch does not happen to mTORC1 signaling or hypoxia or glycolysis.

9. How are the genes shown in Figure 3F chosen for?

10. How are the genes in Figure 4 chosen?

11. Fig 4D: How was this network constructed?

12. In the section "Ependymoma mesenchymal-like tumor cells consist of multiple cell subpopulations with distinct transcriptomic profile and signaling activity": (A) the authors refer to "spectral graph approach" and no additional detail is provided. We understand that the authors refer to an R package in the Methods section, but a motivation or discussion of why this method was chosen would be very helpful. For example, why not simply sub-cluster MLCs? (B) How are the specified genes selected for? (C) Figure 4G: We request the authors to perform comparison at a gene set or signature level as opposed to against one gene. (D) There is no figure provided for "MYC and ATF3 were upregulated in a subpopulation of MLCs with high gene expression levels and binding motif accessibility of JUN". (E) In the methods section, the authors say that they had to subsample the data to 500 cells to get this method to work efficiently. Was the results tested for robustness under different sampling of cells? Why were 500 cells chosen, does the method not work at all for 1000 cells for example? Please provide more details.

13. The authors seem to correlate RNA and ATAC data to some extent in Figure 4. Perhaps a more direct way would be to do a joint analysis. The authors could use Seurat package (for example) to do a joint analysis of the data.

14. We request the authors to make it clear or at least motivate clearly in wording why the authors chose TNFa in the experiment shown in Figure 5.

Minor comments:

1. In the results section the authors refer to 46 tumors while in the Methods section they refer to 44 tumors.

2. In the first section of "Results" the authors write "All tumor-derived cell populations were located adjacent to the cluster of undifferentiated tumor cells". But so does OPC. The authors are requested to comment on it.

3. In the "Bulk Rna-seq processing" in the Methods section, the authors are requested to provide motivation for the calculation done to "assess the importance of each signature".

4. The transition from "Posterior fossa ependymoma tumor cells and infiltrating microglia express pro-inflammatory cues" to "Mesenchymal-like tumor cells are associated with abundant vascularization and microglia infiltration, and have elevated expression of NFkB target genes" is not clear.

5. When authors introduce a new method (for example CIBERSORTx or copysCAT or Laplacian based method), we request them to discuss what the method does in a sentence or two.

6. It is not clear why the authors inferred chromosomal aberration (CA) from ATAC data. What does it contribute to the manuscript?

7. In all figures showing gene expression, the expression scale is not provided. Please provide a color bar. Also, how does this adjust with Harmony batch correction (related to comment 5 in the specific comments section)?

8. Fig. 2A: what does the negative ES look like? Fig. 2B: what is positive and negative in the volcano plot?

9. Fig. 3E: Are these all the TFs that satisfy the criteria? How were they chosen?

10. The authors are requested to provide parameters used for the processing of ATAC-seq data.

Response to the Reviewers' Comments

Comments from Reviewer #1:

“Aubin, et al. reported the mesenchymal transition of ependymoma was induced by the pro-inflammatory cytokines in the tumor microenvironment, by means of single-cell chromatin accessibility analysis and gene expression profiling. They identified NFκB, activator protein-1, and MYC as important mediators of this transition. This is a very interesting paper which discriminates tumor cells from the cells of microenvironment- composing cells such as astrocytes and microglia as the cytokine producing cells for the first time in this field.

For the benefit of the reader, some points need clarifying and certain statements require further justification. These are given below.”

We thank the reviewer for the constructive comments and positive feedback. In the revised manuscript, we have included additional clarifications to support the conclusions of the paper. We describe these changes in what follows:

“1) The authors insist that the source of TGFβ are the mesenchymal-like tumor cells (autocrine) and astroblast (paracrine). If so, what is the first event in the EMT of ependymoma?”

Although the main cause of EMT is hypoxia, ependymoma typically has sparse necrotic areas in the pathological examination. Also, astroblast cannot usually release a large amount of TGFβ.”

From our study we cannot unambiguously identify the first event in the mesenchymal transformation of ependymoma tumor cells. However, our results suggest that any event triggering TGF-β secretion in the tumor and/or microenvironment would lead to the mesenchymal transformation of ependymoma tumor stem cells. In this regard, hypoxia, reactive oxygen species, and ionizing radiation are natural candidates to trigger the mesenchymal transformation of ependymoma, similarly to what has been described in glioblastoma.

In the revised manuscript, we have included a new paragraph in the Discussion section to clarify this point.

“2) As the authors described that the abundance of mesenchymal-like tumor cells was strongly correlated with the abundance of microglia, it is natural to interpret the infiltrated microglia is the main source of TGFβ. Microglia is mesodermal origin and is high producer of TGFβ.”

Our single-cell RNA-seq data is indeed consistent with tumor-infiltrating microglia being an important source of TGF-β1. The following UMAP representation of the microglia cell population shows the expression of the *TGFB1* gene:

TGFB1

We agree with the reviewer that the role of microglia as a non-tumor source of TGF- β 1 was not clearly stated in the original version of the manuscript. In the revised manuscript, we have explicitly mentioned this information in the Results (subheading “Posterior fossa ependymoma tumor cells and infiltrating microglia express pro-inflammatory cues”) and Discussion sections.

“3) Please explain the identification of astroblast in the UMAP of the single-nuclei RNA-seq analyses appeared in Figure 1. The definition of astroblast is not easy. Are these cells positive for GFAP and S-100 protein, or other molecular markers are definitely positive? The origin of astroblast may be the mesenchymal-like tumor cells in this study. There are a subtype of tancytic ependymoma which has features intermediate between astrocytes and ependymal cells. More detailed pathological diagnoses of the original 41 tumors are necessary.”

We agree with the reviewer that there is generally no perfect terminology to refer to tumor-derived cell populations since, although they present similitudes with ordinary cell populations, being transformed cells they also present numerous differences. We denoted the referred cell population as “tumor-derived astroblasts” based on the co-expression of *GFAP*, *S100B*, and *AQP4* in the single-nucleus RNA-seq data, as show in Fig. 1b. In addition, our single-nucleus ATAC-seq data also shows the activity of the transcription factors NFIB and NFIC in this cell population (Fig. 3e), which are implicated in the regulation of astrocytic differentiation (Wilczynska et al. 2009).

There are several observations in our analysis that indicate that this cell population does not originate from the mesenchymal-like tumor cells:

1. The cell differentiation trajectories that result from the RNA velocity (Fig. 1f) do not connect this population with the mesenchymal-like tumor cell population, but with the undifferentiated tumor cell population.
2. This cell population does not express mesenchymal marker genes, such as *VEGFA*, *CHI3L1*, and *CA9* (Figs. 1b and 1c)
3. Our single-nucleus ATAC-seq data shows the activity of neuro-epithelial transcription factors (such as *SOX2*) and the inactivity of mesenchymal transcription factors (such as *RELA*) in this cell population (Fig. 3e).
4. Our analysis of the bulk gene expression data shows no significant correlation between the abundance of mesenchymal-like cells and this cell population (Spearman’s correlation $r = 0.03$, p -value = 0.8).

Note also that we found this cell population in the 9 posterior fossa tumors that we have profiled with single-cell RNA-seq. However, none of the 9 tumors was classified as tancytic

ependymoma according to the clinical information provided by CBTN. To confirm this information, two board-certified neuropathologists in our team (Dr. Santi and Dr. Nasrallah) have independently verified that none of the 9 tumors present histopathological characteristics of tanyctic ependymoma based on histological examination of FFPE tissue sections.

To clarify this point in the revised manuscript, we have provided more details in the Results section (subheading “A single-nucleus transcriptomic atlas of primary and metastatic posterior fossa ependymoma”) about how the 10 sub-populations of Fig. 1a were annotated. We have also renamed the population of “tumor-derived astroblasts” as “tumor-derived astrocytic cells”, as the former term might suggest a connection with astroblastoma which our study has not shown. Finally, we have added the histological type (classic (grade 2) / anaplastic (grade 3)) of each tumor in Supplementary Table 1.

“4) The comparison between the original and metastatic tumors is tricky, because the metastatic tumors usually experienced radiotherapy. Have all the tumors experienced radiotherapy? This point should be described in the Results section and mentioned in the Discussion section.”

The reviewer is right that all the metastatic tumors and the recurrence in our study experienced radiotherapy before surgery. We agree that this is a relevant point, as it has been demonstrated in glioblastoma that ionizing radiation can induce the mesenchymal transformation of the tumor cells (Bhat et al. 2013; Minata et al. 2019). We have specified this information in the Results section (subheading “A single-nucleus transcriptomic atlas of primary and metastatic posterior fossa ependymoma”) of the revised manuscript and discussed the potential implications in the Discussion section.

Comments from Reviewer #2:

“The authors study the gene regulatory circuits in pediatric posterior fossa ependymoma to better understand how tumor cells transition to a mesenchymal-like state. They provide a single-nuclei chromatin accessibility and gene expression atlas of primary and metastatic posterior fossa ependymoma tumors through single RNA seq and ATAC seq data. The authors find the transition of tumor cells into a mesenchymal-like state is characterized by the inactivation of neuroepithelial transcription factors, such as SOX2 and members of the Nuclear Factor I and Regulatory Factor X families, and the activation of transcription factors from the NFkB, AP-1, MAF/BACH, MYC, Hippo, and sonic hedgehog signaling pathways. The mesenchymal-like cell population is characterized by the expression of pro-inflammatory cytokines, hypoxia, angiogenesis, and glycolysis programs.

Overall, the manuscript is novel with results that are noteworthy with strong methodology. The results of the study are significant to the field of neuro-oncology and contributes to the existing literature. One criticism of the approach is only 1 patient tumor was included that was recurrent.”

We thank the reviewer for the constructive comments and positive feedback. We agree that having more than one recurrence would have been desirable in our study. However, given the various constraints (molecular subtype, balanced sex, quality of RNA in the flash-frozen

specimen, etc.) and limited availability of tissue, we were only able to include one recurrence in the study. Nevertheless, our results show that there is a substantial overlap between the initially diagnosed tumors and the metastases, both in terms of the cell populations that are present (although in different proportions), and the gene expression programs of those cell populations. Based on this observation, we believe the conclusions of our study are not substantially affected by only having one recurrence in the single-nucleus data.

“1. The MLC population is characterized by the mesenchymal-like gene signature; however, the authors find this population quite heterogeneous. What percentage of tumor cells are comprised by these tumor cells?”

In the new Supplementary Figure 8 of the revised manuscript, we present the fraction of MLCs comprised by each subpopulation of MLCs:

These cell subpopulations have been annotated based on gene-set enrichment analysis (GSEA) (see also our response to point 12 of reviewer #4). According to the single-nucleus RNA-seq data, the fraction comprised by metabolic MLCs is enriched in the 4 progressive and metastatic samples (fold change = 2.2, Wilcoxon rank-sum test p -value = 0.06). However, as we describe in point 5 below, we are not able to confirm this enrichment in the Heidelberg cohort, possibly due to the small number of metastatic samples in this cohort.

“2. What WHO grade were the tumor studied in both primary, recurrent, and metastatic tumors?”

We have specified the WHO grade of the tumors in the Supplementary Table 1 of the revised manuscript. As expected, most of the tumors in our cohort (7 out of 9) were histologically classified as grade 3 anaplastic ependymoma, whereas the remaining ones (2 out of 9, both being primary tumors) were classified as grade 2 classic ependymoma. Nevertheless, please note that no definitive association between grade and biological behavior or survival has been established for ependymoma, and for these tumors the molecular group has a higher prognostic or predictive value (Louis et al. 2021).

“3. The authors refer to the MLCs as tumor-derived reactive glia, how resistant are these cells to standard therapies such as radiation therapy? Do these cells have gene expression that suggests they have better DNA repair mechanisms to resist radiation therapy treatment? Do

they have expression of genes suggestive of chemotherapy resistance (e.g., ABC transporters, miRNA expression), are resistant to apoptosis, and have hypoxic stability?”

We would like to thank the reviewer for this suggestion. In the revised manuscript we have performed a more in-depth analysis of the expression of genes associated with resistance to chemotherapy and radiation therapy. We find that some of the subpopulations of MLCs that we have identified in our study express high levels of *ABCC3*, which codes for Multidrug Resistance-Associated Protein 3 and is associated with chemotherapeutic resistance in several cancers (Balaji et al. 2016; Zhao et al. 2013), *PDK1*, which leads to HIF-1 α -mediated radio-resistance (Zhao et al. 2017), and *SOD2*, which mediates resistance to radiation through oxidative stress modulation (Fan et al. 2007; Holley et al. 2010). The following figure shows the expression of these genes in the UMAP representation of the single-nucleus RNA-seq data:

The expression of these genes in the larger CBTN and Heidelberg cohorts of tumors profiled at the bulk level confirms these results by showing a strong correlation with the inferred abundance of MLCs in each patient (Spearman's $r = 0.7-0.9$, p -value $< 10^{-10}$). We have presented these new results in the Results section of the revised manuscript (subheading “Ependymoma mesenchymal-like tumor cells consist of multiple cell subpopulations with distinct transcriptomic profile and signaling activity”) and in the new Supplementary Figure 9.

“4. Do the authors believe the tumor-derived reactive glia may represent a type of tumor stem cell that may propagate tumors and make them more aggressive clinically?”

There are several observations that indicate that this might be the case:

1) The expression of cell proliferation markers by tumor-derived reactive glia in our single-nucleus data suggest that this population has proliferative potential:

TOP2A

MKI67

- 2) Our in vitro cell migration experiment indicates that some of the tumor-derived glia have increased migratory potential, which could lead to a more aggressive clinical phenotype.
- 3) Consistent with this observation, the expression of mesenchymal markers in ependymoma has been associated with poor clinical prognosis (Wani et al. 2012; Gillen et al. 2020).
- 4) A similar mesenchymal-like population in glioblastoma has been directly linked to resistance to radiation therapy (Bhat et al. 2013; Minata et al. 2019).

Nevertheless, our work also indicates that tumor-derived reactive glia is a heterogeneous cell population, and it is possible that only a subset is actually associated with a more aggressive clinical phenotype. Further work beyond the scope of our manuscript is needed to clarify this point.

In the revised manuscript, we have added a paragraph in the Discussion section where we discuss the significance and limitations of our results.

“5. A higher motility phenotype is suggested by the MLC phenotype of tumors in the manuscript, were these cells more abundant in the tumor population studied from the metastatic and recurrent tumors?”

Using marker genes of MLC subpopulations as a proxy for their relative abundance in the tumors profiled at the bulk level, we observe an enrichment of metabolic MLCs over angiogenic MLCs in recurrent and metastatic samples in the large CBTN cohort of tumors (e.g. fold change between *PPP1R15A* and *CHI3L1* expression = 5.8, Wilcoxon rank-sum p-value = 0.007, with similar fold-changes for other marker genes). However, we are not able to reproduce this result in the Heidelberg cohort (fold change between *PPP1R15A* and *CHI3L1* expression = 0.94, Wilcoxon rank-sum p-value = 0.4, with similar fold-changes for other marker gene choices), possibly due to the small number of metastatic tumors in this cohort. In general, we think that a future study involving longitudinal data from the same tumors will be needed to address this question in an unambiguous manner, as we describe in our response to point 7. We therefore prefer to be cautious and not present these partial results in the manuscript.

“6. The authors show that the transition of posterior fossa ependymoma tumor cells into a mesenchymal like state involves the inhibition of neuroepithelial transcription factors and the activation of the NFkB and AP-1 complexes. What do the authors propose is the switch that causes this transition of PFA tumor cells?”

Our results suggest that any stimulus or genetic/epigenetic alteration that leads to the persistent activation of these pathways in the neuro-epithelial-like stem cells of the tumor can lead to their transition into a mesenchymal-like state, analogous to what has been observed in glioblastoma. In that regard, we expect that hypoxia, ionizing radiation, inflammation, and reactive oxygen species may favor the mesenchymal transformation of ependymal tumors. We have added some text in the Discussion section of the revised manuscript to clarify this point.

“7. Did the authors find the microglia cell populations to be more abundant within each tumor type studied (primary, recurrent, and metastatic)? Are the tumor associated microglia more abundant within the recurrent and metastatic tumors? Are they responsible for the EMT switch in tumors with their association with MLCs?”

We observe a strong relation between the abundances of microglia and MLCs in the bulk RNA-seq data, for both the CBTN and Heidelberg cohorts, as we describe in the Results section of the manuscript (subheading “Mesenchymal-like tumor cells are associated with abundant vascularization and microglia infiltration and have elevated expression of NFkB target genes”). It has been also established that the expression of mesenchymal markers and the abundance of MLCs in ependymoma is associated with poorer survival (Wani et al. 2012; Gillen et al. 2020). However, our data do not allow us to confidently evaluate changes in microglia abundance along progression due to the small number of recurrences and metastasis in the CBTN and Heidelberg cohorts, and the lack of longitudinal data from the same tumors. In particular, we find that the inferred fold-change in the abundance of microglia between primary tumors and recurrences/metastasis is non-significant and differs between the two cohorts considered in our study (FC = 0.6 for the CBTN and FC = 1.1 for the Heidelberg cohort). In addition, the comparison based on bulk data might be sensitive to several covariates: 1) the tumor purity of a sample will likely diminish as the tumor progresses and becomes more diffuse, 2) the abundance of microglia may be affected by the very different physiological conditions at the loci of distal metastases (e.g. spine vs hindbrain), and 3) complex scenarios where the cells seeding distal metastases are derived from MLCs induced by tumor-infiltrating microglia in the primary tumor, but the metastases themselves do not have a high abundance of microglia, are not ruled out. In general, as mentioned above, we think that a future study involving a larger number of metastatic samples and longitudinal data from the same tumors will be needed to address this question in an unambiguous manner. In the revised manuscript, we have explicitly stated this limitation and the need for large longitudinal studies of ependymoma to directly observe the evolution of microglia and MLC subpopulations during tumor progression.

“Are the microglia recruited from the circulation in PFA tumors?”

Please note that distinguishing resident microglia from bone-marrow-derived macrophages in the human brain based on gene expression is challenging and to the best of our understanding still an open question in the field (Buonfiglioli and Hambardzumyan 2021). A recent study has proposed a set of microglia-specific and bone-marrow-derived macrophage-specific gene expression markers in the murine central nervous system (Haage et al. 2019). Our single-nucleus RNA-seq data shows high gene expression of those microglia-specific markers in the tumor-infiltrating microglia population, and low gene expression of macrophage-specific markers:

However, although these results are interesting, to the best of our knowledge the validity of these markers in the context of human glioma needs to be further established.

We have presented these results and caveats in the Results section (subheading “Posterior fossa ependymoma tumor cells and infiltrating microglia express pro-inflammatory cues”) and the new Supplementary Fig. 2 of the revised manuscript.

“8. Could the microglia account for a more invasive phenotype associated with the MLCs?”

We think this is a plausible hypothesis based on 1) the expression of TGF- β 1 in tumor infiltrating microglia, 2) the migratory phenotype induced by TGF- β 1 in vitro, and 3) the known association of MLCs with poor clinical prognosis (Wani et al. 2012; Gillen et al. 2020). However, due to the limitations stated above (see our response to points 5 and 7), we think that longitudinal data from the same patients will be needed to fully clarify this point.

In the Discussion section of the revised manuscript, we have stated this hypothesis and the current limitations of our study.

Comments from Reviewer #3:

“In the manuscript “Pro-Inflammatory Cytokines Mediate the Epithelial-to-Mesenchymal-Like Transition of Pediatric Posterior Fossa Ependymoma” Rachael G. Aubin et al. performed single cell chromatin accessibility and transcriptomic profiling of primary ependymomas in the posterior fossa and metastasis in spine and cortex to reveal transcription factors and enhancers associated with differentiation of tumor cells to amongst other mesenchymal-like cells (MLC). Using ligand-receptor pair analysis from snRNA-seq they infer proinflammatory signaling between microglia and MLC and show in a patient derived cell model that cells with mesenchymal gene expression signature show diverse phenotypes depending on stimulation with TGF β and/or TNF α .”

I have several major concerns regarding the study, which are listed below:”

We thank the reviewer for all the constructive comments, which we have addressed in the revised manuscript as we detail in what follows:

“1) Please specify how sample were selected to be included in snRNA-seq and snATAC-seq analysis? The number of biosamples for single cell analysis is comparably low and the samples are spread across many sub-types (PFA-1, PFA-2, PFB) and stages (initial, progressive, and distal metastasis). Please specify what PFB and PFA correspond to and how they relate to PFA-1 and 2.”*

Samples were selected based on the following criteria:

- Age at the time of surgery < 16 years old.
- Location in the posterior fossa (if primary tumor or recurrence) or derived from primary tumor located in the posterior fossa (if distal metastasis).
- No other brain surgery during the previous 8 months.
- Enough tissue available in the biorepository (>100 mg, so that our study does not substantially impacts the stock of the biobank).
- Well-preserved RNA in the tissue (as determined with a bio-analyzer).

We have added a sentence in the Methods section of the revised manuscript (subheading “Tumor samples”) specifying this information. We have also defined the PFB and PFA* acronyms in the legend of Supplementary Table 1, where they are used.

We did not limit our study to PFA-1 tumors because the same tumor-derived cell populations are observed (in different proportions) across PFA-1, PFA-2, and PFB tumors, and the transcriptome of these molecular groups appears to form a continuum (Gillen et al. 2020). Additionally, a substantial amount of heterogeneity is present within the PFA-1 and PFA-2 subtypes, as it has been demonstrated in (Pajtler et al. 2018). Therefore, even restricting to PFA tumors would lead to a substantial amount of heterogeneity. More importantly, the goal of our study was not to find differences between molecular sub-types, but to characterize the mechanisms of mesenchymal transformation of posterior fossa ependymoma that are present across specific sub-types. In this regard, as detailed in our response to point 5 of reviewer #4, we have verified that none of the results reported in our study is driven by a few patients or molecular sub-type.

“Validation experiments performed in cell model from a cell line derived from a recurrent PFA ependymoma. Given the tumor heterogeneity and different trajectories cells might take I was wondering what cell types/stage does EPD-210FHTC correspond to or is it a mixture?”

The EPD-210FHTC cell model has been characterized and utilized as a model of PFA ependymoma in multiple publications (Brabetz et al. 2018; Pajtler et al. 2018; Gojo et al. 2020; Panwalkar et al. 2021). It has been profiled with single-cell RNA-seq in (Gojo et al. 2020), where it was shown (see Figure S1H of that reference) that it consists of a mixture of the same tumor cell populations observed in the patients, with the mesenchymal cell population (denoted as “PF-Metabolic” in that figure) representing a very small fraction of the cells at baseline.

“2) Sample M9 (and same is true to lesser extent for M7) which according to snRNA-seq data consists of 60% of MLC (Fig. 1E) and thus has the highest fraction of MLCs, but is listed somewhere in the middle in the analysis of inferred fraction of MLC from bulk RNA-seq (2C) and does not support the pathways hypoxia, glycolysis etc which the authors associate with the MLC population. Indeed this sample seems depleted for TNF α signaling through NF κ B. This puts into question the conclusions related to the gene programs active in MLCs. Do the other samples to the left in 2C have indeed a higher fraction of MLCs? Based on Figure 1E this seems at least unlikely since all L_ samples have <5% of MLCs. How do the authors reconcile this discrepancy?”

We thank the reviewer for this comment and agree that additional clarification was needed in the manuscript. Ependymomas have a substantial amount of spatial heterogeneity (Gillen et al. 2020). In particular, MLCs are localized in perinecrotic and perivascular regions. For example, the following images from Fig. 4c of (Gillen et al. 2020) show the immunostaining of 3 posterior fossa tumors for CA9, a marker of MLC according also to our single-nucleus data.

The scale bar in these images is 2 mm. The volume of the tissue samples that we used to generate our single-nucleus data is approximately $\sim 8 \text{ mm}^3$. Thus, the cell proportions in the single-nucleus data are not necessarily representative of the overall proportions across the entire tumor. We stated this in the Results section of the original manuscript (subheading “Single cell chromatin accessibility profiling enables the systematic study of primary and metastatic posterior fossa ependymoma gene regulatory circuits”), when comparing the cell abundances observed in the single-nucleus ATAC-seq and single-nucleus RNA-seq data of the same tumors: “The abundance of each cell population in each tumor was moderately correlated with the observed abundance of the same population in the single-nucleus RNA-seq data of the

same tumor (Figs. 3B and 1E, Pearson's $r = 0.56$, p -value $< 10^{-4}$), likely due to the large degree of spatial heterogeneity of posterior fossa ependymoma (Gillen et al. 2020)". In the revised manuscript, we have added a more detailed explanation in the Results section (subheading "A single-nucleus transcriptomic atlas of primary and metastatic posterior fossa ependymoma"). The cell proportions inferred from the bulk RNA-seq data should better represent the overall proportions, as they are based on larger tumor samples, but they should be still subjected to a substantial amount of variability.

However, note that the pathway enrichment analyses of the single-nucleus RNA-seq data are only based on the gene expression profile of MLCs that are present in the tumor sample and do not depend on the cell abundances in the single-nucleus data. Similarly, the pathway enrichment analyses of the bulk data we have performed show the correlation between the enrichment for a set of pathways in a sample of the tumor and the abundance of MLCs in that specific sample, providing support for the gene expression programs that are active in the MLCs. We have verified these associations in 4 independent ways and datasets: gene expression in the single-nucleus RNA-seq data, chromatin accessibility in the single-nucleus ATAC-seq data, gene expression in the CBTN cohort, and gene expression in the Heidelberg cohort. We thus do not expect our results to be affected in any substantial way by the large-scale spatial variability of ependymal tumors.

"3) RNA velocity analysis is based on comparing spliced and unspliced RNA molecules and it seems there is no consensus if the assumptions and models are applicable to profiling of single nuclei. Thus, it is unclear what to make out of Fig 1F and if the indicated relationships and trajectories between cell types can be concluded from the data itself even if in agreement with the literature."

We agree with the reviewer that RNA velocity was initially conceived for single-cell RNA-seq data. However, we and others have noted a large abundance of spliced transcripts in single-nucleus RNA-seq data. To the best of our knowledge, the dynamic model of scVelo can be applied to single-nucleus data if the degradation rate is reinterpreted as the nuclear export rate. Several published papers have performed RNA velocity analyses on single-nucleus RNA-seq data (see for example, Fig. 3C of (Marsh and Bleloch 2020) and Figs. 2O-S of (Wen et al. 2021)). Please note that the trajectories that resulted from our analysis are consistent with those obtained from fresh tumor tissue samples (Gojo et al. 2020). In addition, a trajectory analysis of our data using Monocle3 also leads to consistent results, as it can be seen in the following figure showing the UMAP representation and differentiation trajectories produced by Monocle3, where cells have been colored according to the same cell identities presented in the manuscript:

We agree with the reviewer that these assumptions need to be mentioned in the manuscript. For that purpose, we have added a sentence in the Methods section of the revised manuscript (subheading “RNA velocity field”) specifying that to apply the dynamic model of scVelo to single-nucleus RNA-seq data we reinterpreted the RNA degradation rate as the nuclear export rate.

“4) From the presented data it is difficult to follow the proposed EMT-like process because the MLC cluster consists of >90% cells from the 2 metastasis samples. Thus, it is possible that cells changed their phenotype and expression profile in response to the microenvironment or prior to migration or these are cells of different origin rather than transitioned. Here it could be interesting to look within one patient into both tumor and metastasis or compare MLC within a tumor sample to the corresponding other clusters while excluding MLCs from metastases samples. To illustrate that these are derived from tumor cells the ATAC data with inferred copy number variations could be helpful, since MLCs (if transitioned) would have the same CNVs as the origin cell. According to 3D both tumors with chrom. aberrations had no MLC. A snATAC-seq dataset of samples with MLC and chrom aberrations might help to address this.”

We would like to thank the reviewer for this suggestion. Please note that although the MLC cluster consists of >90% cells from metastatic samples, all of the samples, including also the primary tumors, have MLCs in our dataset. In particular, the MLCs in the primary tumor I4 share the same chromosomal aberrations than the other tumor cells, as shown in Supplementary Fig. 7 of the revised manuscript, indicating a common origin:

This is consistent with what has been described for the mesenchymal transformation of other gliomas, where the transition between neuroepithelial (usually denoted as pro-neural in glioblastoma) and mesenchymal tumor stem cells is dynamic and bidirectional (Kim et al. 2021).

As suggested by the reviewer, we have reprocessed, integrated, and performed an RNA velocity analysis of the primary tumors alone, without including the recurrence and the metastatic samples. We have visualized the trajectories that result from this analysis in the same UMAP representation of Fig. 1a to facilitate the comparison:

As it can be seen in this figure, the resulting trajectories from the analysis of primary tumors are consistent with those of the combined primary, recurrent, and metastatic tumors presented in Fig. 1f of the manuscript. In particular, a trajectory from undifferentiated tumor cells into MLCs is still present in the analysis of primary tumors.

Altogether these results support to the interpretation presented in the manuscript where MLCs originate from undifferentiated neuroepithelial tumor cells. In the Results section of the revised manuscript (subheading “Single cell chromatin accessibility profiling enables the systematic

study of primary and metastatic posterior fossa ependymoma gene regulatory circuits”), we have explicitly mentioned that MLCs share the same CAs than the other tumor cell populations in each patient, suggesting a common origin.

“5) Please explain why TFs differentially expressed between NE and MLC were used to infer EMT regulators? Based on the trajectory displayed there is no direct relationship between NE and MLC. Why not compare to undiff. cells or any of the other cell types?”

Please note that we use the term “neuro-epithelial tumor cells” to denote all the tumor cells that are not mesenchymal (MLCs). It therefore comprises undifferentiated, ependymal, NPCs, and astrocytic cell populations. We define this term in the Results section (subheading “A single-nucleus transcriptomic atlas of primary and metastatic posterior fossa ependymoma”). The RNA velocity field of Figure 1f shows a trajectory from some of the neuroepithelial cells (predominantly from the undifferentiated cells) into the MLCs. The differential expression analysis presented in the manuscript is based on the comparison of MLCs with the neuroepithelial cell populations collectively. Nevertheless, restricting this analysis to undifferentiated neuroepithelial cells instead of all neuroepithelial cells does not substantially change the results and the same set of TFs are significant.

“6) Fig. 3A how were NPC separated from undifferentiated cells by the indicated dotted line? Fig 3B there is a huge increase in the fraction of MLCs in I2 and I5 > 5% in RNA to ~40% in the ATAC dataset. Was a very different region of the tumor used or is this linked to lower resolution and thus separation of clusters in ATAC? Since MLCs are a major focus of this study these drastic differences in composition should be addressed, e.g. by 2 technical repeats from a tumor.”

We identified the population of NPCs in the ATAC-seq data by noticing the localization of the activity of neurogenic transcription factors, such as ASCL1 and NHLH1, within the cluster of undifferentiated tumor cells / NPCs. The degree of localization of the transcription factor activity was statistically assessed using the Laplacian score (Govek, Yamajala, and Camara 2019). In the revised manuscript, we have instead adopted a more conventional approach and subclustered the population of undifferentiated tumor cells / NPCs to separate the population of NPCs into a discrete cluster:

We have updated Figure 3 accordingly in the revised manuscript.

We have addressed the mismatch in the fraction of MLCs between different samples of the same tumor in our response to point 2. As we discuss in detail there, ependymal tumors involve a substantial amount of spatial heterogeneity which leads to differences in the cell proportions present in different samples of the same tumor. However, none of the analyses in our manuscript is expected to depend on this variability.

“7) Please provide metadata for each single cell experiments such as how many cells per data set, genes and transcripts/nucleus for RNA, fragments/nucleus and a measure for signal-to-noise ratio for snATAC-seq such as TSS enrichment.”

We have specified the metadata in the Results section of the revised manuscript (subheadings “A single-nucleus transcriptomic atlas of primary and metastatic posterior fossa ependymoma” and “Single cell chromatin accessibility profiling enables the systematic study of primary and metastatic posterior fossa ependymoma gene regulatory circuits”). Our single-nucleus RNA-seq data consists of 25,349 nuclei (2,660 nuclei per sample on average, with a mean of 544 detected genes per nucleus). Our single-nucleus ATAC-seq data consists of 14,461 nuclei (mean number of nuclei per sample = 2,410) and 229,286 accessible peaks (mean number of fragments per nucleus = 8,122, mean transcription start site enrichment score = 4.5). Although the number of detected genes per nucleus in the single-nucleus RNA-seq data is low compared to single-cell RNA-seq datasets from fresh tissue, it is comparable to what is often observed in single-nucleus RNA-seq data from flash-frozen archived tumor samples (for example, see metadata for ovarian cancer and CLL single-nucleus datasets in figure 5b of (Slyper et al. 2020)). On the other hand, the ability to profile flash-frozen archived samples by means of single-nucleus RNA-seq provides a unique tool for the study of pediatric cancers, where the incidence is small and it is difficult to procure enough tissue (in the case of pediatric posterior fossa ependymoma, ~150 new cases per year in the US).

“Minor Point:

8) Please show marker genes for undifferentiated tumor cells as well (1B)? How were tumor from non-tumor cells distinguished in snRNA-seq data?”

Please note that there are no differentially expressed genes in the undifferentiated tumor cell population (Supplementary Table 2). All the genes expressed by this population were also expressed in other tumor cell populations at high levels. Based on that observation we labeled this cell population as “undifferentiated tumor cells”. We agree with the reviewer that this needed more clarity in the manuscript and have updated the Results section (subheading “A single-nucleus transcriptomic atlas of primary and metastatic posterior fossa ependymoma”).

The distinction between tumor and non-tumor cells in the single-nucleus RNA-seq data was based on the expression of tumor genes, such as *GRIA4*, *CFAP157*, and *CD44*, and the continuity of these populations in the gene expression space. More importantly, the annotations were confirmed by the copy number aberrations inferred for each cell population in the single-nucleus ATAC-seq data (Supplementary Fig. 7 of the revised manuscript).

“9) Fig. 1E and Fig. 3B stack plot to display contributions of samples to different clusters would allow for easier interpretation.”

Thank you for the suggestion. We have replaced the bar plots by stacked bar charts in Figures 1e and 3b. We have also included a new Supplementary Fig. 4 with stacked bar charts representing the inferred cell population abundances for the CBTN and Heidelberg bulk-level cohorts.

Comments from Reviewer #4:

“Aubin et al. characterize the neuroepithelial-like tumor cells and mesenchymal-like tumor cells at the single-cell level using RNA and ATAC modalities in their manuscript titled "Pro-Inflammatory Cytokines Mediate the Epithelial-to-Mesenchymal-Like Transition of Pediatric Posterior Fossa Ependymoma". The authors investigate gene-regulatory mechanisms that lead to EMT-like phenotype in ependymoma and seek to understand the relationship of this process with inflammation. While the work has potential to be impactful, the manuscript itself has several shortcomings and need to be addressed.

Generally, a lot of the examples in the manuscript (for example choice of genes to investigate) seem to be chosen without providing any data-driven motivation. This also extends to computational methods where a lot of details are missing.

More specific comments:”

We thank the reviewer for the positive assessment and all the constructive suggestions, which we have addressed in the revised manuscript as we detail in what follows:

“1. In the introduction, the authors suggest that prior studies (Gillen et al. 2020, Gojo et al. 2020, Griesinger et al. 2015, Griesinger et al. 2017) have already described EMT-like process that involves NFkB activation and hypoxia that aids growth and propagation of the tumors. How the results of the authors differentiate from the prior studies is not revisited at the conclusion of the manuscript. We encourage authors to clearly differentiate their work.”

We would like to thank the reviewer for this suggestion. Previous single-cell RNA-seq studies of posterior fossa ependymoma have been limited to just indicating the presence of a cell population with a mesenchymal-like gene expression signature (Gillen et al. 2020; Gojo et al. 2020). Our work largely expands the results of these studies by providing a more detailed and mechanistic understanding of the mesenchymal-like cell population of posterior fossa ependymoma. This is in part due to the large abundance of this cell population in our data, the incorporation of single-nucleus ATAC-seq data in addition to RNA-seq data, and the experiments using a patient-derived cell model treated with cytokines. Thus, we have identified the main transcription factors and regulatory elements involved in the mesenchymal transformation of ependymoma and provided multiple lines of evidence for the involvement of the inflammatory tumor microenvironment in this process. We have identified specific cytokines involved in the mesenchymal transformation and provided some experimental validation using a patient-derived cell model. In addition, we have shown that the mesenchymal gene-expression signature comprises a heterogeneous population of cells consistent with distinct functionalities (metabolic, angiogenic, etc.).

In the revised manuscript, we have edited the Discussion section to more clearly differentiate our work from previous works.

“2. How were the 10 sub-populations identified in Figure 1A annotated? Is the heatmap in Fig 1B showing the top differentially expressed genes? And was this heatmap used to annotate the clusters? Or was it the other way around? We suggest the authors to clearly state how the obtained clusters were annotated as specific cell types? It is also not clear how the undifferentiated cells were identified.”

The 10 sub-populations identified in Figure 1a were annotated based on the results of the differential expression analysis presented in Supplementary Table 2. In particular, we used known markers of cell types that were differentially expressed, as well as enrichments for gene sets associated with specific cell types.

The identity of the population of undifferentiated tumor cells was assigned based on the lack of differentially expressed genes in this cell population and the observation that the genes expressed by this cell population were also expressed in other tumor cell populations.

To clarify this point in the revised manuscript, we have indicated in the Results section (subheading “A single-nucleus transcriptomic atlas of primary and metastatic posterior fossa ependymoma”) the specific marker genes and pathway enrichments that were used to annotate each of the tumor cell populations. We have also indicated in the legend of Figure 1b that the heatmap shows differentially expressed genes that are discussed in the main text because of being markers that we used to annotate the cell populations or because of their relevant biology.

We have also indicated in bold in that figure the genes that were used to annotate cell populations.

“3. (A) The authors use scVelo to construct RNA velocity trajectories. The authors are requested to comment on the time frame of tumor progression and the time frame on which the assumptions made in scVelo operate (that of unspliced/spliced turnover); and to interpret scVelo results accordingly when applied to transition that operates at a different time scale.”

We expect the timeframe of the trajectories inferred by scVelo to be of the order of a few hours, as determined by the rate of transcription and nuclear export. The fact that the trajectories capture the transition between neuro-epithelial and mesenchymal-like tumor stem cells indicates that this transition can occur relatively quickly. This is consistent with what has been described in glioblastoma, where the transition between pro-neural and mesenchymal cell states takes place in a few hours upon treatment with ionizing radiation (see e.g. Fig. 3E of (Minata et al. 2019)). Consistent with those results, treating the patient-derived cell model of our manuscript with TGF- β 1 induces noticeable changes in the expression of *VEGFA* (an early MLC gene) in as little as 6 hours:

Altogether these results indicate that the transition between neuro-epithelial and mesenchymal like cells can occur relatively quickly. This timeframe is of course much smaller than that of tumor progression (typically months) and suggest that additional processes, such as the emergence of genetic or epigenetic alterations that lead to the stabilization and accumulation of this cell population, are needed for this cellular process to have a macroscopic effect in tumor progression.

To clarify this point in the revised manuscript, we have added some comments about the timeframe of the scVelo trajectories in the Discussion section.

“(B) The authors use a different set of parameters to run scVelo as opposed to UMAP/Clustering. We request the authors to motivate this.”

We have verified that using the same set of parameters that we used for UMAP/Clustering ($n_top_genes = 5000$, $n_pcs = 30$, $n_neighbors = 30$) leads to a velocity map that is consistent with the one presented in Figure 1f of the manuscript:

However, we have a slight preference for the parameters used in the manuscript as they do not lead to an overall directionality for the IPC sub-population.

We have added a sentence in the Methods section of the revised manuscript (subheading “RNA velocity field”) stating the consistency of the RNA velocity map with other parameter choices.

“4. At the end of the first section in "Results" the authors claim "numerous genes coding for components of the WNT signaling pathway were upregulated in non-mesenchymal" and present a few example genes. Why were only these specific genes chosen? The authors are requested to perform GSEA and show that WNT pathway is enriched here. Or at least show that genes of WNT pathway have a high expression using a proper gene signature as opposed to a set of few genes.”

We would like to thank the reviewer for this suggestion. In the revised manuscript we have performed GSEA for the WNT pathway in the non-mesenchymal tumor cells, showing a significant enrichment (normalized enrichment score = 3.05, p -value = 0.04). We have presented the results of this analysis in the Results section of the revised manuscript (subheading “A single-nucleus transcriptomic atlas of primary and metastatic posterior fossa ependymoma”).

“5. The authors seem to directly apply batch correction method. However, it would be more informative if they could provide reasoning for this. Perhaps, show visual of the data before and after the correction and argue why batch correction needs to be done. Tumors have biological variance between patients, how do the authors know that the variance they see (if they do) between batches is not biological? Perhaps the authors can highlight non-tumor cells and use it as a guide to show effect of batch correction (ideally, non-tumor cells like immune cells should be similar in phenotype across patients). Also, the authors use Harmony to correct batch effect, but Harmony does not correct for gene expression. How do the authors ensure their downstream calculations are not impacted by this? Is this done by computing differential expression in each sample separately? We request the authors to be explicit about this.

The comment on batch effect correction also applies for ATAC-seq data.”

We agree with the reviewer that in general there is a substantial amount of biological variability between tumor cells of different patients, in addition to technical variability. As the reviewer mentions, this can be observed in a UMAP representation of the non-corrected single-nucleus RNA-seq data:

As it can be seen in this representation, non-tumor cells have a more consistent gene expression profile across patients than tumor cells, suggesting a smaller biological variability. A similar situation is also observed in the non-corrected representation of the single-nucleus ATAC-seq data:

Since in our study we are not interested in biological differences between individual patients, but on cellular phenotypes that are consistent across patients, the use of a batch correction method such as Harmony is well motivated.

We only used the consolidated latent space to cluster the cells into cell populations. To ensure that downstream analyses like differential gene expression analysis are not driven by individual samples, we performed those analyses individually for each sample, and combined the p-values using Fisher's method, as described in the Methods section of the manuscript (subheadings "Differential gene expression analysis", "Differentially accessible cis-regulatory elements", and "Motif enrichment analysis").

To clarify this point in the revised manuscript, we have included both the non-corrected and corrected representations of the single-nucleus RNA and ATAC-seq data in the Supplementary Figure 1. In addition, we have explicitly mentioned in the Results section (subheading “A single-nucleus transcriptomic atlas of primary and metastatic posterior fossa ependymoma”) the motivation for this approach.

“6. In the analysis using CellChat, how are the pathways identified? Is there any quantification to rank them? And how are the genes discussed chosen for? Also, CCL seems to be higher from microglia -> MLCs. Is this consistent with the authors' claims? This section generally needs more details and the conclusion made is not clear in how it follows from the data shown.”

CellChat provides a curated database of ligand-receptor/co-receptor pairs that are categorized into signaling pathways based on the KEGG database (Jin et al. 2021). It assigns a p-value to each inferred interaction based on the differential expression of genes coding for ligands and receptors/co-receptors. The complete list of significant interactions and their p-values is presented in Supplementary Table 3. The interactions discussed in the manuscript and summarized in Supplementary Fig. 3 were selected based on their statistical significance (p -value < 0.05) and involvement in cytokine, chemokine, or growth factor signaling between different tumor cell populations or between tumor cell and microenvironment cell populations. As shown in Supplementary Table 3 (tab “Signaling Pathways”) and represented in Supplementary Fig. 3, our results are consistent with the MLCs being the main source of CC chemokines in these tumors. The expression of CC chemokine receptors in our data is generally low, likely due to the lack of sensitivity of single-nucleus RNA-seq. However, the statistical significance of microglia to MLC CCL signaling is lower than that of MLC to microglia CCL signaling (p -values 0.004 and < 0.0004, respectively). This is consistent with an extensive body of literature showing that microglia are the main cell population responding to CC chemokines in gliomas (e.g. (Platten et al. 2003)).

In the revised manuscript, we have extended this part of the Results section (subheading “Posterior fossa ependymoma tumor cells and infiltrating microglia express pro-inflammatory cues”) and the legends of Supplementary Figure 3 and Supplementary Table 3 to provide more details about the CellChat analysis.

“7. Details on how the genes (HIF3A and HIF1A) are selected for in the section: “Mesenchymal-like tumor cells are associated with abundant vascularization and microglia infiltration, and have elevated expression of NFκB target genes” are needed.”

HIF3A is a negative regulator of the transcription factor HIF-1 α , which mediates glycolytic and other major cellular responses to decreased oxygen levels (e.g. as described in (Dengler, Galbraith, and Espinosa 2014)). The observed switch in gene expression is consistent with HIF3A being a key element in the maintenance of the neuroepithelial phenotype. We have added some sentences in the Results section of the revised manuscript (subheading “Mesenchymal-like tumor cells are associated with abundant vascularization and microglia infiltration, and have elevated expression of NF κ B target genes”) to clarify this point.

“8. At the end of the section "Mesenchymal-like tumor cells are associated with abundant vascularization and microglia infiltration, and have elevated expression of NFkB target genes", the authors are requested to show the switch in expression at a signature level as opposed to select genes. Also, the authors are requested to comment on why the switch does not happen to mTORC1 signaling or hypoxia or glycolysis.”

Although GSEA confirms a strong enrichment for the sonic hedgehog gene expression signature in samples with a high abundance of mesenchymal-like cells (Fig 2c, Spearman's correlation between the GSEA normalized score and the abundance of MLCs $r = 0.5$, FDR < 0.01), we have not been able to confirm an enrichment for genes involved in the positive regulation of WNT signaling in samples with a high abundance of neuroepithelial cells (Spearman's correlation between the GSEA normalized score of WNT signature genes and the abundance of MLCs $r = -0.2$, p -value = 0.2). In the revised manuscript, we therefore no longer talk in this section about a switch between Sonic hedgehog and WNT signaling, but only about the activation of Sonic hedgehog signaling (as well as mTORC1, hypoxia, glycolysis, etc.) in MLCs.

“9. How are the genes shown in Figure 3F chosen for?”

Fig. 3f shows the motif accessibility score of 5 TFs that are significantly associated with specific tumor-derived cell populations and that are discussed in the main text because of their relevant biology. The complete list of transcription factors significantly associated with tumor cell populations is presented in Supplementary Table 5.

We discuss MEIS1 and MEIS2 in the text because they are the only transcription factors that we find associated with both undifferentiated tumor cells and mesenchymal cells. As we mention in the text, these are potential targets for effective therapies, their role in tumor progression has been documented in other cancers, and small-molecule inhibitors are being developed (Bhanvadia et al. 2018; Cheng et al. 2021; Yao et al. 2021). NHLH1 is a canonical pro-neural basic helix-loop-helix TF (Dennis, Han, and Schuurmans 2019). Its association (as well as that of ASCL1) with the NPC population provides further support to the annotation of this population as NPCs. The activity of the glucocorticoid and mineralocorticoid receptors NR3C1/2 uniquely marks the population of tumor-derived astrocytic cells. RFX transcription factors regulate motile ciliogenesis (Choksi et al. 2014). Their association with the population of tumor-derived ependymal cells therefore provides additional support for the identity of that population.

To clarify this point in the revised manuscript, we have provided additional context for the relevance of these transcription factors in the Results section (subheading “Single cell chromatin accessibility profiling enables the systematic study of primary and metastatic posterior fossa ependymoma gene regulatory circuits”).

“10. How are the genes in Figure 4 chosen?”

We have repeated this analysis in a more systematic manner. In Figure 4a of the revised manuscript we present all the transcription factors that have differentially accessible binding motifs across the EMT-like process (Wilcoxon rank-sum test, FDR < 0.01) and for which their gene expression is significantly correlated or anti-correlated (Spearman correlation p -value <

0.05) with the abundance of MLCs in the CBTN and Heidelberg bulk RNA-seq cohorts. We have also made this clearer in the legends of Figure 4a and Supplementary Table 7 of the revised manuscript.

“11. Fig 4D: How was this network constructed?”

We have added a subheading “Regulatory network between transcription factors with differentially accessible binding motifs across the EMT-like process” in the Methods section of the revised manuscript describing how this network was constructed. In brief, we downloaded the binding sites of members of the MAF/BACH, NF κ B, and AP-1 complexes with differentially accessible binding motifs across the EMT-like process from the JASPAR2020 database and overlaid the location of differentially accessible peaks (as described in “Differentially accessible of cis-regulatory elements” of the Methods section). The network was then built by taking the transcription as nodes and adding a directed edge from one TF into another if one or more significant (FDR < 0.05) differentially accessible binding sites of the first transcription factor were present in the gene locus (\pm 10 kbp upstream and downstream) of the second transcription factor.

“12. In the section "Ependymoma mesenchymal-like tumor cells consist of multiple cell subpopulations with distinct transcriptomic profile and signaling activity": (A) the authors refer to "spectral graph approach" and no additional detail is provided. We understand that the authors refer to an R package in the Methods section, but a motivation or discussion of why this method was chosen would be very helpful. For example, why not simply sub-cluster MLCs? (B) How are the specified genes selected for? (C) Figure 4G: We request the authors to perform comparison at a gene set or signature level as opposed to against one gene. (D) There is no figure provided for "MYC and ATF3 were upregulated in a subpopulation of MLCs with high gene expression levels and binding motif accessibility of JUN".”

MLCs do not separate into distinct clusters, probably indicating that different MLCs molecular phenotypes emerge through dynamic and continuous cellular processes. Clustering methods can be relatively unstable in those situations. Because of that, we used the Laplacian score for feature selection to identify genes that are differentially expressed in some MLC subpopulation, without having to pre-specify subpopulations. More details about this type of analysis can be found in (Govek, Yamajala, and Camara 2019) (section “Differential expression analysis”). The genes represented in Figure 4f are significant genes under this analysis (FDR < 0.01) that display a tendency towards mutual exclusivity.

For consistency, in the revised manuscript we have also included a more standard analysis based on sub-clustering the MLCs (see also our response to point 1 of reviewer #2). The results of this analysis are presented in Supplementary Figure 8. In this new analysis, we have annotated MLC subpopulations based on gene-set enrichment analysis and not on the expression of individual marker genes, as described in the Results section (subheading “Ependymoma mesenchymal-like tumor cells consist of multiple cell subpopulations with distinct transcriptomic profile and signaling activity”) of the revised manuscript. We have removed the reference to the binding motif accessibility of JUN in the revised manuscript.

“(E) In the methods section, the authors say that they had to subsample the data to 500 cells to get this method to work efficiently. Was the results tested for robustness under different sampling of cells? Why were 500 cells chosen, does the method not work at all for 1000 cells for example? Please provide more details.”

Computing the statistical significance of the Laplacian score is computationally costly. For that reason, we subsampled cell populations to 500 cells when computing the Laplacian score in the original version of the manuscript. In the revised version of the manuscript, we have increased this number to 1,000 cells. We have verified the consistency of the results across different sets of randomly sampled cells. The following plot shows the Pearson’s correlation coefficient between the Laplacian scores of each gene for 5 different repetitions of this analysis on the MLC population:

In all cases, the pairwise Pearson’s correlation coefficient is >0.9 , and the p-value is $< 10^{-16}$, demonstrating a large degree of stability of the results against different sets of randomly sampled cells.

To clarify this point in the revised manuscript, we have added a sentence in the Methods section (subheading “Differential gene expression analysis”).

“13. The authors seem to correlate RNA and ATAC data to some extent in Figure 4. Perhaps a more direct way would be to do a joint analysis. The authors could use Seurat package (for example) to do a joint analysis of the data.”

We agree with the reviewer that integrating the RNA and ATAC-seq data into a single latent space based on the gene activity scores inferred from the ATAC-seq data can in some cases be an efficient approach. However, in our study this approach is limited by two types of covariates (data modality and tumor Id), as well as by the substantial differences between the single-nucleus ATAC-seq and RNA-seq data. The following UMAP representation shows the result of integrating the single-nucleus ATAC-seq and RNA-seq datasets using Seurat4:

As it can be seen in this representation, the single-nucleus RNA-seq and ATAC-seq datasets are still substantially localized in different regions of the integrated representation. Moreover, although cells belonging to the same cell type tend to localize in the same region of the representation,

cells from different patients also separate in the combined representation:

Based on these limitations we have therefore decided not to include this analysis in the revised manuscript.

“14. We request the authors to make it clear or at least motivate clearly in wording why the authors chose TNF α in the experiment shown in Figure 5.”

We included TNF- α in the experiment of Figure 5 because the gene expression and chromatin accessibility data presented in Figures 2 and 4 show that NF κ B activation is one of the main characteristics of MLCs. TNF- α is a major activator of the NF κ B pathway (Hayden and Ghosh 2014) and our analysis of candidate ligand-receptor interactions using the single-nucleus RNA-seq data is consistent with tumor-infiltrating microglia being a source of TNF- α signaling for the tumor cell populations (Figure 1g).

In the Results section of the revised manuscript (subheading “TGF- β 1 and TNF- α respectively induce and modulate the EMT in a patient derived PFA cell model”), we have more clearly motivated in wording the choice of TNF- α for these experiments.

“Minor comments:

1. In the results section the authors refer to 46 tumors while in the Methods section they refer to 44 tumors.”

We would like to thank the reviewer for this comment. We have fixed this in the Results section (subheading “A single-nucleus transcriptomic atlas of primary and metastatic posterior fossa ependymoma”) of the revised manuscript.

“2. In the first section of "Results" the authors write "All tumor-derived cell populations were located adjacent to the cluster of undifferentiated tumor cells". But so does OPC. The authors are requested to comment on it.”

We agree with the reviewer that this population appears also relatively adjacent to the cluster of undifferentiated tumor cells in the UMAP representation of Figure 1. However, this changes substantially across different analyses. For instance, in the UMAP representation produced by Monocle (see our response to point 3 of reviewer #3), OPCs appear clearly separated from the

undifferentiated tumor cells. Similarly, the OPC cluster appears clearly separated from the tumor cells in the single-nucleus ATAC-seq data (Figure 3a). More importantly, the analysis of chromosomal aberrations inferred from the single-nucleus ATAC-seq data (Supplementary Figure 7) shows the absence of chromosomal aberrations in this cell population, consistently with being mostly composed of non-transformed cells.

We have added 2 sentences clarifying this point in the Results section (subheading “A single-nucleus transcriptomic atlas of primary and metastatic posterior fossa ependymoma”) of the revised manuscript.

“3. In the "Bulk Rna-seq processing" in the Methods section, the authors are requested to provide motivation for the calculation done to "assess the importance of each signature".”

We have reworded this part in the Methods section of the revised manuscript to improve the clarity of the presentation.

“4. The transition from "Posterior fossa ependymoma tumor cells and infiltrating microglia express pro- inflammatory cues" to "Mesenchymal-like tumor cells are associated with abundant vascularization and microglia infiltration, and have elevated expression of NFkB target genes" is not clear.”

We have reworded this transition in the revised manuscript to improve the clarity of the presentation.

“5. When authors introduce a new method (for example CIBERSORTx or copyScAT or Laplacian based method), we request them to discuss what the method does in a sentence or two.”

We thank the reviewer for this suggestion and have implemented this in the revised manuscript.

“6. It is not clear why the authors inferred chromosomal aberration (CA) from ATAC data. What does it contribute to the manuscript?”

The main purpose of this analysis was to provide further support to our annotation of tumor vs non-tumor cell populations. We have made this clearer in the revised manuscript.

“7. In all figures showing gene expression, the expression scale is not provided. Please provide a color bar. Also, how does this adjust with Harmony batch correction (related to comment 5 in the specific comments section)?”

We have included color expression scales in all figures. The units are $\log_2(1+100 \text{ CPM})$, so expression values are adjusted for differences in library size. Note that Harmony batch correction does not adjust the gene expression levels. As mentioned in our response to major point 5 above, to ensure that downstream analyses like differential gene expression analysis are not driven by individual samples, we performed those analyses individually for each sample, and combined the p-values using Fisher’s method.

“8. Fig. 2A: what does the negative ES look like? Fig. 2B: what is positive and negative in the volcano plot?”

We have indicated in Figure 2 of the revised manuscript the interpretation of positive and negative scores.

“9. Fig. 3E: Are these all the TFs that satisfy the criteria? How were they chosen?”

The updated figure shows all the TFs that have a significant binding motif accessibility score in at least one of the tumor cell populations with an FDR < 0.01 and an average fold-change of one standard deviation or more ($\Delta z \geq 1$). The complete list of TFs with a significant motif accessibility score (FDR < 0.1) is presented in Supplementary Table 5. We have clarified this in the legends of Figure 3e and Supplementary Table 5 of the revised manuscript.

“10. The authors are requested to provide parameters used for the processing of ATAC-seq data.”

We have provided that information in the Methods section of the revised manuscript (subheading “Single-nucleus ATAC-seq data processing”). In brief, we keep cells based on the number (>1,500 and <35,000) and percentage (>20%) of fragments in peaks, the ratio of mononucleosomal to nucleosome-free fragments (<3), and the transcription start site (TSS) enrichment score (>2), the percentage of mitochondrial fragments (<10%) and the percentage of fragments overlapping targeted sites (<20% or 25%, depending on the sample).

References

- Balaji, S. A., N. Udupa, M. R. Chamallamudi, V. Gupta, and A. Rangarajan. 2016. 'Role of the Drug Transporter ABCC3 in Breast Cancer Chemoresistance', *PLoS One*, 11: e0155013.
- Bhanvadia, R. R., C. VanOpstall, H. Brechka, N. S. Barashi, M. Gillard, E. M. McAuley, J. M. Vasquez, G. Paner, W. C. Chan, J. Andrade, A. M. De Marzo, M. Han, R. Z. Szmulewitz, and D. J. Vander Griend. 2018. 'MEIS1 and MEIS2 Expression and Prostate Cancer Progression: A Role For HOXB13 Binding Partners in Metastatic Disease', *Clin Cancer Res*, 24: 3668-80.
- Bhat, K. P. L., V. Balasubramanian, B. Vaillant, R. Ezhilarasan, K. Hummelink, F. Hollingsworth, K. Wani, L. Heathcock, J. D. James, L. D. Goodman, S. Conroy, L. Long, N. Lelic, S. Wang, J. Gumin, D. Raj, Y. Kodama, A. Raghunathan, A. Olar, K. Joshi, C. E. Pelloski, A. Heimberger, S. H. Kim, D. P. Cahill, G. Rao, W. F. A. Den Dunnen, Hwgm Boddeke, H. S. Phillips, I. Nakano, F. F. Lang, H. Colman, E. P. Sulman, and K. Aldape. 2013. 'Mesenchymal differentiation mediated by NF-kappaB promotes radiation resistance in glioblastoma', *Cancer Cell*, 24: 331-46.
- Brabetz, S., S. E. S. Leary, S. N. Grobner, M. W. Nakamoto, H. Seker-Cin, E. J. Girard, B. Cole, A. D. Strand, K. L. Bloom, V. Hovestadt, N. L. Mack, F. Pakiam, B. Schwalm, A. Korshunov, G. P. Balasubramanian, P. A. Northcott, K. D. Pedro, J. Dey, S. Hansen, S. Ditzler, P. Lichter, L. Chavez, D. T. W. Jones, J. Koster, S. M. Pfister, M. Kool, and J. M. Olson. 2018. 'A biobank of patient-derived pediatric brain tumor models', *Nat Med*, 24: 1752-61.
- Buonfiglioli, A., and D. Hambardzumyan. 2021. 'Macrophages and microglia: the cerberus of glioblastoma', *Acta Neuropathol Commun*, 9: 54.

- Cheng, M., Y. Zeng, T. Zhang, M. Xu, Z. Li, and Y. Wu. 2021. 'Transcription Factor ELF1 Activates MEIS1 Transcription and Then Regulates the GFI1/FBW7 Axis to Promote the Development of Glioma', *Mol Ther Nucleic Acids*, 23: 418-30.
- Choksi, S. P., G. Lauter, P. Swoboda, and S. Roy. 2014. 'Switching on cilia: transcriptional networks regulating ciliogenesis', *Development*, 141: 1427-41.
- Dengler, V. L., M. Galbraith, and J. M. Espinosa. 2014. 'Transcriptional regulation by hypoxia inducible factors', *Crit Rev Biochem Mol Biol*, 49: 1-15.
- Dennis, D. J., S. Han, and C. Schuurmans. 2019. 'bHLH transcription factors in neural development, disease, and reprogramming', *Brain Res*, 1705: 48-65.
- Fan, M., K. M. Ahmed, M. C. Coleman, D. R. Spitz, and J. J. Li. 2007. 'Nuclear factor-kappaB and manganese superoxide dismutase mediate adaptive radioresistance in low-dose irradiated mouse skin epithelial cells', *Cancer Res*, 67: 3220-8.
- Gillen, A. E., K. A. Riemony, V. Amani, A. M. Griesinger, A. Gilani, S. Venkataraman, K. Madhavan, E. Prince, B. Sanford, T. C. Hankinson, M. H. Handler, R. Vibhakar, K. L. Jones, S. Mitra, J. R. Hesselberth, N. K. Foreman, and A. M. Donson. 2020. 'Single-Cell RNA Sequencing of Childhood Ependymoma Reveals Neoplastic Cell Subpopulations That Impact Molecular Classification and Etiology', *Cell Rep*, 32: 108023.
- Gojo, J., B. Engliger, L. Jiang, J. M. Hubner, M. L. Shaw, O. A. Hack, S. Madlener, D. Kirchhofer, I. Liu, J. Pyrdol, V. Hovestadt, E. Mazzola, N. D. Mathewson, M. Trissal, D. Lotsch, C. Dorfer, C. Haberler, A. Halfmann, L. Mayr, A. Peyrl, R. Geyeregger, B. Schwalm, M. Mauermann, K. W. Pajtler, T. Milde, M. E. Shore, J. E. Geduldig, K. Pelton, T. Czech, O. Ashenberg, K. W. Wucherpennig, O. Rozenblatt-Rosen, S. Alexandrescu, K. L. Ligon, S. M. Pfister, A. Regev, I. Slavc, W. Berger, M. L. Suva, M. Kool, and M. G. Filbin. 2020. 'Single-Cell RNA-Seq Reveals Cellular Hierarchies and Impaired Developmental Trajectories in Pediatric Ependymoma', *Cancer Cell*, 38: 44-59 e9.
- Govek, K. W., V. S. Yamajala, and P. G. Camara. 2019. 'Clustering-independent analysis of genomic data using spectral simplicial theory', *PLoS Comput Biol*, 15: e1007509.
- Haage, V., M. Semtner, R. O. Vidal, D. P. Hernandez, W. W. Pong, Z. Chen, D. Hambardzumyan, V. Magrini, A. Ly, J. Walker, E. Mardis, P. Mertins, S. Sauer, H. Kettenmann, and D. H. Gutmann. 2019. 'Comprehensive gene expression meta-analysis identifies signature genes that distinguish microglia from peripheral monocytes/macrophages in health and glioma', *Acta Neuropathol Commun*, 7: 20.
- Hayden, M. S., and S. Ghosh. 2014. 'Regulation of NF-kappaB by TNF family cytokines', *Semin Immunol*, 26: 253-66.
- Holley, A. K., Y. Xu, D. K. St Clair, and W. H. St Clair. 2010. 'RelB regulates manganese superoxide dismutase gene and resistance to ionizing radiation of prostate cancer cells', *Ann N Y Acad Sci*, 1201: 129-36.
- Jin, S., C. F. Guerrero-Juarez, L. Zhang, I. Chang, R. Ramos, C. H. Kuan, P. Myung, M. V. Plikus, and Q. Nie. 2021. 'Inference and analysis of cell-cell communication using CellChat', *Nat Commun*, 12: 1088.
- Kim, Y., F. S. Varn, S. H. Park, B. W. Yoon, H. R. Park, C. Lee, R. G. W. Verhaak, and S. H. Paek. 2021. 'Perspective of mesenchymal transformation in glioblastoma', *Acta Neuropathol Commun*, 9: 50.
- Louis, D. N., A. Perry, P. Wesseling, D. J. Brat, I. A. Cree, D. Figarella-Branger, C. Hawkins, H. K. Ng, S. M. Pfister, G. Reifenberger, R. Soffietti, A. von Deimling, and D. W. Ellison. 2021. 'The 2021 WHO Classification of Tumors of the Central Nervous System: a summary', *Neuro Oncol*, 23: 1231-51.
- Marsh, B., and R. Blelloch. 2020. 'Single nuclei RNA-seq of mouse placental labyrinth development', *Elife*, 9.
- Minata, M., A. Audia, J. Shi, S. Lu, J. Bernstock, M. S. Pavlyukov, A. Das, S. H. Kim, Y. J. Shin, Y. Lee, H. Koo, K. Snigdha, I. Waghmare, X. Guo, A. Mohyeldin, D. Gallego-Perez, J. Wang, D. Chen, P. Cheng, F. Mukheef, M. Contreras, J. F. Reyes, B. Vaillant, E. P. Sulman, S. Y. Cheng, J. M. Markert,

- B. A. Tannous, X. Lu, M. Kango-Singh, L. J. Lee, D. H. Nam, I. Nakano, and K. P. Bhat. 2019. 'Phenotypic Plasticity of Invasive Edge Glioma Stem-like Cells in Response to Ionizing Radiation', *Cell Rep*, 26: 1893-905 e7.
- Pajtler, K. W., J. Wen, M. Sill, T. Lin, W. Orisme, B. Tang, J. M. Hubner, V. Ramaswamy, S. Jia, J. D. Dalton, K. Hauptfear, H. A. Rogers, C. Punchihewa, R. Lee, J. Easton, G. Wu, T. A. Ritzmann, R. Chapman, L. Chavez, F. A. Boop, P. Klimo, N. D. Sabin, R. Ogg, S. C. Mack, B. D. Freibaum, H. J. Kim, H. Witt, D. T. W. Jones, B. Vo, A. Gajjar, S. Pounds, A. Onar-Thomas, M. F. Roussel, J. Zhang, J. P. Taylor, T. E. Merchant, R. Grundy, R. G. Tatevossian, M. D. Taylor, S. M. Pfister, A. Korshunov, M. Kool, and D. W. Ellison. 2018. 'Molecular heterogeneity and CXorf67 alterations in posterior fossa group A (PFA) ependymomas', *Acta Neuropathol*, 136: 211-26.
- Panwalkar, P., B. Tamrazi, D. Dang, C. Chung, S. Sweha, S. K. Natarajan, M. Pun, J. Bayliss, M. P. Ogrodzinski, D. Pratt, B. Mullan, D. Hawes, F. Yang, C. Lu, B. R. Sabari, A. Achreja, J. Heon, O. Animasahun, M. Cieslik, C. Dunham, S. Yip, J. Hukin, J. J. Phillips, M. Bornhorst, A. M. Griesinger, A. M. Donson, N. K. Foreman, H. J. L. Garton, J. Heth, K. Muraszko, J. Nazarian, C. Koschmann, L. Jiang, M. G. Filbin, D. Nagraath, M. Kool, A. Korshunov, S. M. Pfister, R. J. Gilbertson, C. D. Allis, A. M. Chinnaiyan, S. Y. Lunt, S. Bluml, A. R. Judkins, and S. Venneti. 2021. 'Targeting integrated epigenetic and metabolic pathways in lethal childhood PFA ependymomas', *Sci Transl Med*, 13: eabc0497.
- Platten, M., A. Kretz, U. Naumann, S. Aulwurm, K. Egashira, S. Isenmann, and M. Weller. 2003. 'Monocyte chemoattractant protein-1 increases microglial infiltration and aggressiveness of gliomas', *Ann Neurol*, 54: 388-92.
- Slyper, M., C. B. M. Porter, O. Ashenberg, J. Waldman, E. Drokhlyansky, I. Wakiro, C. Smillie, G. Smith-Rosario, J. Wu, D. Dionne, S. Vigneau, J. Jane-Valbuena, T. L. Tickle, S. Napolitano, M. J. Su, A. G. Patel, A. Karlstrom, S. Gritsch, M. Nomura, A. Waghray, S. H. Gohil, A. M. Tsankov, L. Jerby-Arnon, O. Cohen, J. Klughammer, Y. Rosen, J. Gould, L. Nguyen, M. Hofree, P. J. Tramontozzi, B. Li, C. J. Wu, B. Izar, R. Haq, F. S. Hodi, C. H. Yoon, A. N. Hata, S. J. Baker, M. L. Suva, R. Bueno, E. H. Stover, M. R. Clay, M. A. Dyer, N. B. Collins, U. A. Matulonis, N. Wagle, B. E. Johnson, A. Rotem, O. Rozenblatt-Rosen, and A. Regev. 2020. 'A single-cell and single-nucleus RNA-Seq toolbox for fresh and frozen human tumors', *Nat Med*, 26: 792-802.
- Wani, K., T. S. Armstrong, E. Vera-Bolanos, A. Raghunathan, D. Ellison, R. Gilbertson, B. Vaillant, S. Goldman, R. J. Packer, M. Fouladi, I. Pollack, T. Mikkelsen, M. Prados, A. Omuro, R. Soffietti, A. Ledoux, C. Wilson, L. Long, M. R. Gilbert, K. Aldape, and Network Collaborative Ependymoma Research. 2012. 'A prognostic gene expression signature in infratentorial ependymoma', *Acta Neuropathol*, 123: 727-38.
- Wen, Y., D. A. Englund, B. D. Peck, K. A. Murach, J. J. McCarthy, and C. A. Peterson. 2021. 'Myonuclear transcriptional dynamics in response to exercise following satellite cell depletion', *iScience*, 24: 102838.
- Wilczynska, K. M., S. K. Singh, B. Adams, L. Bryan, R. R. Rao, K. Valerie, S. Wright, I. Griswold-Prenner, and T. Kordula. 2009. 'Nuclear factor I isoforms regulate gene expression during the differentiation of human neural progenitors to astrocytes', *Stem Cells*, 27: 1173-81.
- Yao, M., Y. Gu, Z. Yang, K. Zhong, and Z. Chen. 2021. 'MEIS1 and its potential as a cancer therapeutic target (Review)', *Int J Mol Med*, 48.
- Zhao, H., H. Jiang, Z. Li, Y. Zhuang, Y. Liu, S. Zhou, Y. Xiao, C. Xie, F. Zhou, and Y. Zhou. 2017. '2-Methoxyestradiol enhances radiosensitivity in radioresistant melanoma MDA-MB-435R cells by regulating glycolysis via HIF-1alpha/PDK1 axis', *Int J Oncol*, 50: 1531-40.
- Zhao, Y., H. Lu, A. Yan, Y. Yang, Q. Meng, L. Sun, H. Pang, C. Li, X. Dong, and L. Cai. 2013. 'ABCC3 as a marker for multidrug resistance in non-small cell lung cancer', *Sci Rep*, 3: 3120.

Reviewers' Comments:

Reviewer #1:

Remarks to the Author:

Thank you for your polite reply. This study uses a powerful technique and will contribute to the understanding of this tumor. However, there is one point I would like you to consider again.

As I mentioned in the first review, ependymoma usually does not have a large necrotic area like glioblastoma. And, radiation therapy is not given at the stage of tumor resection. It is not necessarily appropriate to apply the cause of EMT in glioblastoma to that in ependymoma. For ependymoma, other causes should be considered in the discussion section. Those would include an inflammation or metabolic burden unique to ependymal cells, specific reasons for the recruitment of microglia. It does not, of course, deny the causes of EMT in glioblastoma such as hypoxia.

That should lead to a better understanding of this tumor and a new therapeutic approach.

Reviewer #2:

Remarks to the Author:

The authors have responded well to the critiques.

Reviewer #3:

Remarks to the Author:

Thank you for addressing my comments in the revision.

I have only minor additional request:

Please define the term "neuro-epithelial tumor cells" in the results section. I was not able to find the exact term (only "tumor cells) in the indicated results section (subheading "A single-nucleus transcriptomic atlas of primary and metastatic posterior fossa ependymoma"), only in the figure legend.

It is not clear to me what the labelling "astrocytic cells" exactly means? I would suggest labelling either astrocyte or astrocyte-like cells. For the reader, it would also be helpful to define in the text how e.g. tumor derived astrocytes were distinguished from normal astrocytes.

Reviewer #4:

Remarks to the Author:

The authors have address the reviewer's concerns. We recommend the paper for publication.

Response to the Reviewers' Comments

Comments from Reviewer #1:

"Thank you for your polite reply. This study uses a powerful technique and will contribute to the understanding of this tumor. However, there is one point I would like you to consider again.

As I mentioned in the first review, ependymoma usually does not have a large necrotic area like glioblastoma. And, radiation therapy is not given at the stage of tumor resection. It is not necessarily appropriate to apply the cause of EMT in glioblastoma to that in ependymoma. For ependymoma, other causes should be considered in the discussion section. Those would include an inflammation or metabolic burden unique to ependymal cells, specific reasons for the recruitment of microglia. It does not, of course, deny the causes of EMT in glioblastoma such as hypoxia.

That should lead to a better understanding of this tumor and a new therapeutic approach."

We thank the reviewer for the positive feedback and constructive comment. We have modified the Discussion section to better contextualize the potential causes of the EMT in ependymoma.

Comments from Reviewer #3:

"Thank you for addressing my comments in the revision.

I have only minor additional request:

Please define the term "neuro-epithelial tumor cells" in the results section. I was not able to find the exact term (only "tumor cells) in the indicated results section (subheading "A single-nucleus transcriptomic atlas of primary and metastatic posterior fossa ependymoma"), only in the figure legend.

It is not clear to me what the labelling "astrocytic cells" exactly means? I would suggest labelling either astrocyte or astrocyte-like cells. For the reader, it would also be helpful to define in the text how e.g. tumor derived astrocytes were distinguished from normal astrocytes."

We thank the reviewer for these suggestions. In this revision, we have clearly defined the term "neuroepithelial tumor cells" in the Results section (subheading "A single-nucleus transcriptomic atlas of primary and metastatic posterior fossa ependymoma") and denoted the astrocytic population as "astrocyte-like cells".